# Molecular programming modulates hepatic lipid metabolism and adult metabolic risk in the offspring of obese mothers in a sex-specific manner

Christina Savva[1,2], Luisa A. Helguero [3], Marcela González-Granillo[2], Tânia Melo[4,5], Daniela Couto [4,5], Bo Angelin [1,2], Maria Rosário Domingues[4,5], Xidan Li[1], Claudia Kutter [6] & Marion Korach-André[1,7 ✉]

Male and female offspring of obese mothers are known to differ extensively in their metabolic adaptation and later development of complications. We investigate the sex-dependent responses in obese offspring mice with maternal obesity, focusing on changes in liver glucose and lipid metabolism. Here we show that maternal obesity prior to and during gestation leads to hepatic steatosis and inflammation in male offspring, while female offspring are protected. Females from obese mothers display important changes in hepatic transcriptional activity and triglycerides profile which may prevent the damaging effects of maternal obesity compared to males. These differences are sustained later in life, resulting in a better metabolic balance in female offspring. In conclusion, sex and maternal obesity drive differently transcriptional and posttranscriptional regulation of major metabolic processes in offspring liver, explaining the sexual dimorphism in obesity-associated metabolic risk.

[1] Department of Medicine, Cardiometabolic Unit and Integrated Cardio Metabolic Center, Karolinska Institute, Stockholm, Sweden. [2] Clinical Department of Endocrinology, Karolinska University Hospital Huddinge, Stockholm, Sweden. [3] Institute of Biomedicine, Department of Medical Sciences, University of Aveiro, Aveiro, Portugal. [4] Mass Spectrometry Centre, Department of Chemistry, University of Aveiro, Aveiro, Portugal. [5] CESAM, Centre for Environmental and Marine Studies, Department of Chemistry, University of Aveiro, Aveiro, Portugal. [6] Department of Microbiology, Tumor and Cell Biology, Science for Life Laboratory, Karolinska Institute, Stockholm, Sweden. [7] Department of Gene Technology, Science for Life Laboratory, Royal Institute of Technology (KTH), Stockholm, Sweden. ✉email: marion.korach-andre@scilifelab.se

The alarming increased prevalence of overweight and obese women in reproductive age has urged the need to investigate the impact on fetal health that may become evident later in life. Recent studies have demonstrated strong responses of the offspring to external factors, including nutritional, environmental, and hormonal changes during the prenatal and postnatal periods[1]. Both in human and animal models, embryos exposed to overnutrition during gestation and lactation show metabolic alterations later in life, including increased risk of obesity[2,3], impaired insulin sensitivity and glucose tolerance[4], and increased risk of developing fatty liver disease and hepatocellularcarcinoma[5,6]. Therefore, understanding how maternal diet influences offspring health is of great importance for our ability to better anticipate public health needs, and to develop practices regarding the implementation of dietary and lifestyle interventions.

Important biological and physiological differences have been observed between females and males. These differences are manifested through the sex-biased incidence of many common health problems, including cardiovascular[7], liver[8,9], endocrine, and immune diseases[10]. Recent studies have demonstrated that female and male sex hormones, as well as sex chromosomes, contribute to the development of obesity and insulin resistance[11,12]. Moreover, the development of age-associated diseases mostly occurs in a sex-specific manner, partly correlated with changes in sex hormone levels[12].

Our recent study demonstrated that even when offspring received a control diet after weaning, maternal obesity (MO) altered the hepatic and adipose lipidome of the offspring in a sex-specific manner[13]. Furthermore, sex-specific responses to high calorie-diets have also been described[13–15], implying that sex hormones might play a major role, although the underlying mechanisms are not well understood. Using *ob/ob* mice, we identified sex-specific lipid synthesis pathways in the liver and in the adipose depots which determine the molecular lipid composition, and hereby may play a key role in obesity-associated metabolic risk[16]. Interestingly, estrogen can rescue some of these affected pathways in males by controlling key genes of the lipid synthesis pathways through interaction with the nuclear estrogen receptors alpha and beta[16].

While sex-dependent metabolic adaptation in response to MO has been described[17], the mechanisms by which MO might differently program hepatic lipidome and transcriptional activity in female and male offspring have not been assessed. Therefore, we explored how MO affects adiposity, metabolism, and the hepatic lipidome in obese female and male offspring. First, we examined the sex-specific metabolic profile in high-fat diet offspring, and second, whether MO affected the hepatic lipidome and transcriptome differently in female and male offspring from weaning to 6 months of age. We further evaluated if the maternal and offspring high-fat diet may determine adiposity and liver steatosis in the same individual at different time-point in life (3-months and 6-months) using magnetic resonance imaging and localized spectroscopy. We discovered that the metabolic response to MO is sex-dependent due to sex-specific transcriptional activity of the metabolic pathways in the liver. In males, MO increased hepatic triglyceride (TG) accumulation, reduced mitochondrial activity, and induced inflammatory pathways. On the contrary in females, MO provoked TG remodeling, enhanced mitochondrial activity, and reduced inflammatory pathways. Finally, we also identified sex-specific hepatic lipid molecular species and transcriptional regulations associated with offspring metabolic dysfunctions in obesity.

## Results

**Maternal obesity alters the metabolic profiles in female and male offspring differently.** F0 dam were fed the high-fat (HFD) or the control diet 6 weeks prior mating and remained on their respective diet during pregnancy and lactation (Fig. 1a). Consumption of the HFD by F0 dam for 6 weeks prior to mating led to a significant increase of body weight compared to the control diet-fed F0 dam but had no effect on glucose levels (Fig. 1b), which indicates balanced metabolic processes despite the increased body weight in mothers HFD (moHF). All F1 female and male offspring received the HFD after weaning. Female offspring born from obese mothers (F-moHF) and those born from lean mothers (F-moC) had similar body weight (BW). In contrast, males born from obese mothers (M-moHF) showed significantly lower BW than those born from lean mothers (M-moC) from weaning until week 9 of age and thereafter gained BW to a comparable level of the M-moC. Males weighed significantly more than females after week 10, regardless of the maternal diet (Fig. 1c), with higher food intake (Fig. S1a). Interestingly, food intake tended to be induced in female and reduced in male by MO. To determine if MO altered the adiposity in offspring in the short or/and long term, we defined the body fat distribution by magnetic resonance imaging in the same individual at 12-week (midterm, MID) and 26-week (endterm, END) of age. At MID, M-moHF had lower ratio of total fat on BW than F-moHF due to reduction of fat mass compared to M-moC. At END, F-moC had higher ratio of total fat on BW than M-moC (Fig. 1d). Distribution of visceral and subcutaneous adipose tissue was diet- and sex-dependent. Males had ~11% more visceral and ~9% less subcutaneous fat than females regardless of the maternal diet and timepoint (Fig. 1e, f). Importantly, the ratio between subcutaneous and visceral fat was diet- and sex-dependent (Fig. 1g).

Changes in body fat distribution are closely correlated to metabolic disturbances. At the two timepoints, we evaluated the glucose tolerance and insulin sensitivity in the offspring by oral glucose tolerance and insulin tolerance tests, respectively. Males showed reduced glucose tolerance and insulin sensitivity (high fasting insulin level and during glucose tolerance test) compared to females. However, only females showed reduced insulin sensitivity by MO during the glucose tolerance test (higher insulin levels) (Fig. 1h–k). Both at MID and END, systemic insulin release during fasting and after the glucose load was impaired by MO in females (Fig. 1i–k). Interestingly, insulin tolerance test showed that MO did not alter glucose disappearance in F-moHF (Fig. 1l, m). The quantitative insulin-sensitivity check index indicated that males were more insulin resistant than females at both MID and END, regardless of maternal diet (Fig. 1n). We tested whether the physiological changes described above (in vivo data) were accompanied by changes in gene expression in liver. We performed RNA-seq and considered significantly expressed genes and pathways with FDR < 0.1 and *p*-value <0.05. UMAP plot revealed a clear sex and maternal diet effect on the hepatic transcriptional activity (Fig. 1o). To examine the sex- and diet-dependent changes, we grouped genes that were significantly differentially expressed (DE) between sexes in moC and moHF and by MO in females and males into Kyoto Encyclopedia of Genes and Genomes pathways (Supplementary Data 1). Although we observed different sensitivity to insulin between sexes (Fig. 1i and l), the regulation of insulin signaling pathways was not significantly different between sexes in both diet groups (Fig. 1p; left panel). In contrast, several pathways were regulated differently between sexes by MO (Fig. 1p; right panel). This suggested that MO primes insulin signaling pathways inversely in male and female offspring. We inspected DE genes within insulin signaling pathways (Table S1). The four DE genes *Pdk1*, *Lpin1*, *Prlr*, and *Nox4* are key regulators of hepatic insulin sensitivity[18–21], and were altered by sex and/or by MO. In moC, the expression of *Pdk1*, *Lpin1*, and *Prlr* genes was higher in females than in males. MO reduced the *Pdk1* and *Lpin1* expression in females and induced the *Lpin1* in males. *Nox4* expression was also highly sex-specific with higher expression level in males than in

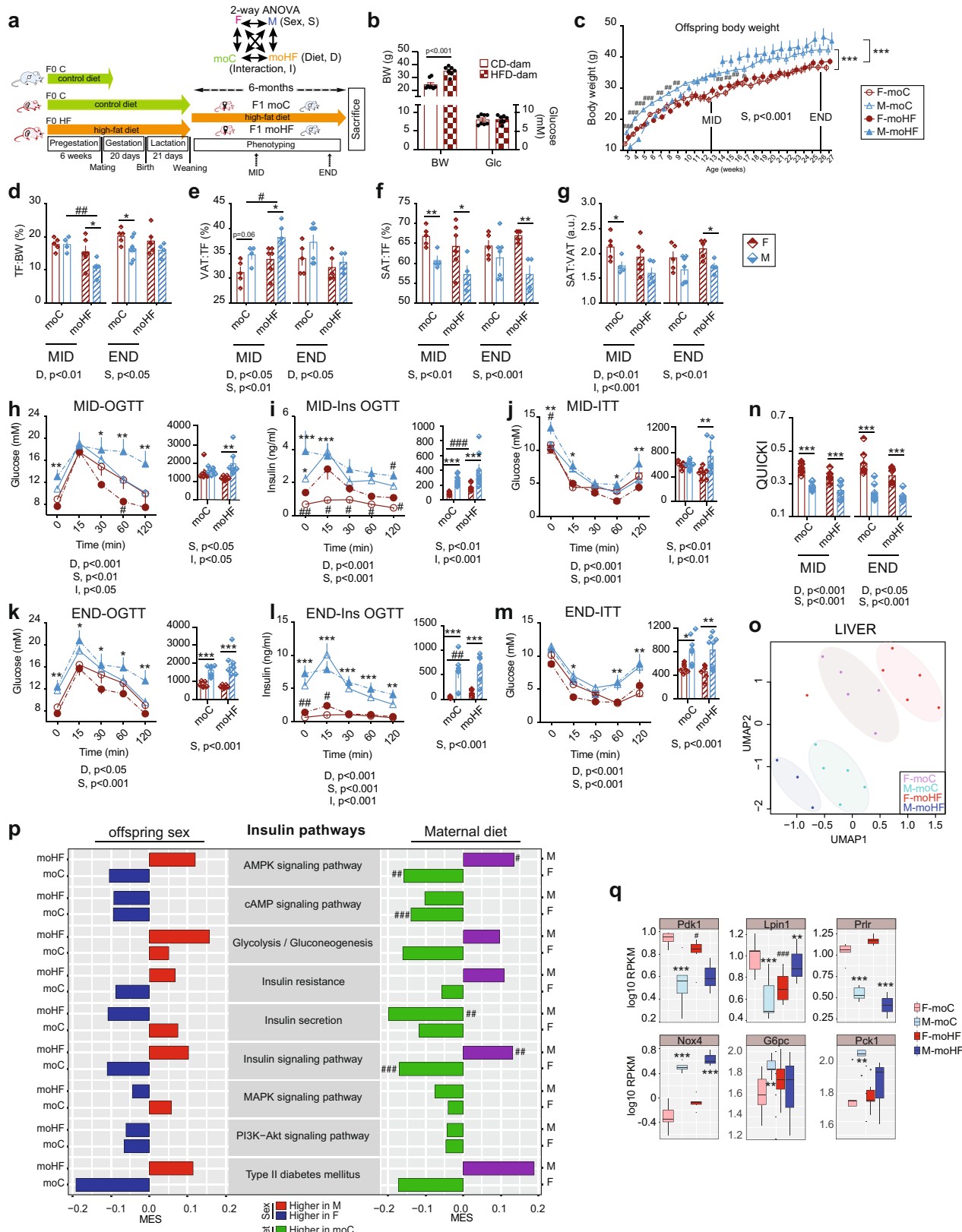

females (Fig. 1q). Moreover, the expression of *G6pc* and *Pck1* involved in gluconeogenesis was higher in M-moC than in F-moC (Fig. 1q). Interestingly, hepatic estrogen receptor alfa (ERS1) has been shown to regulate *G6pc*[22] and *Pck1*[23] and to be critical in the regulation of glucose metabolism, and *Esr1* expression was significantly higher in females than in males in both diet conditions

(Fig. S1b). In addition, a large set of genes was differently regulated between sexes in moC but much less in moHF due to the remodeling of gene activity from insulin pathways in females only (Table S1).

In sum, females but not males showed reduced insulin-controlled circulating glucose level with MO. However, females

**Fig. 1 Physiological and transcriptional response to maternal diet in obese F1 offspring is sex-dependent. a** Graphic description of the study set-up. Dam-F0 were fed either the control diet (C, green arrow) or the high-fat diet (HF, orange arrow) for 6 weeks prior mating and continued the same diet during gestation and lactation; male-F0 remained on C diet until mating. Both female and male F1 offspring remained on HF diet after weaning until sacrifice. The offspring physiological status was assessed in vivo at 3 months (MID) and 6 months (END) of age, using each animal as its own control. Explanatory scheme of the two-way ANOVA statistical comparisons. **b** Dam body weight (BW) and peripheral glucose level. **c** Time series plot of BW in female (F, red circle; open circle in moC and full circle in moHF) and male (M, blue triangle; open triangle in moC and full triangle in moHF) offspring until sacrifice; Bar graphs of the **d** total fat (TF) on BW, **e** visceral adipose tissue (VAT) on TF, **f** subcutaneous adipose tissue (SAT) on TF, and **g** SAT:VAT ratio in F-moC (red open bars), M-moC (blue open bars), F-moHF (red stripped bars), and M-moHF (blue stripped bars) based on MRI images analysis.
**h**–**m** Time-course of the circulating glucose levels and the corresponding insulin levels during the oral glucose tolerance test (OGTT) at, **h** and **i** MID, and **j** and **k** END together with the area under the curve (AUC); glucose levels after insulin injection at **l** MID and **m** END together with the AUC. **n** Quantitative insulin-sensitivity check index (QUICKI) in F-moC (red open bars), M-moC (blue open bars), F-moHF (red stripped bars), and M-moHF (blue stripped bars); **o** UMAP plot of the RNAseq data in offspring's liver. **p** Bar plot presenting the Maximum Estimate Score (MES) between sexes in moC and moHF offspring (left panel), and in response to MO in F and M (right panel) of the selected KEGG pathways involved in insulin and glucose metabolism; red and blue bars indicate higher expression in males and females, respectively, and green and purple bars indicate higher expression in moC and moHF, respectively. **q** Box plots of the expression level (RPKM, $\log_{10}$) of genes of the insulin pathways. For **b**, F-moC ($n = 11$), M-moC ($n = 13$), F-moHF ($n = 11$), and M-moHF ($n = 10$). For **c**–**g**, F-moC ($n = 7$), M-moC ($n = 6$), F-moHF ($n = 7$) and M-moHF ($n = 7$). For **h**–**n**, F-moC ($n = 8$), M-moC ($n = 9$), F-moHF ($n = 7$), and M-moHF ($n = 6$). For **o**–**q**, F-moC ($n = 5$), M-moC ($n = 5$), F-moHF ($n = 6$), and M-moHF ($n = 3$). For **b**–**m** data are presented as mean ± sem; for **q** data are presented as mean ± sd. For **c**–**n**, two-way ANOVA (sex (S), mother diet (D), and interaction (I) between sex and diet followed by Tukey's multiple comparisons test when significant ($p < 0.05$). For **a** and **q**, differences between two groups (sexes, F versus M; maternal diet, moC versus moHF) were determined by unpaired two-tailed $t$-test corrected for multiple comparisons using the Holm–Sidak method, with alpha = 5.000%. For pathway and DE genes analysis we used the Benjamini–Hochberg correction with false Discovery Rate (FDR) values <0.1 when significant. *, M versus F and #, moHF versus moC, $p < 0.05$; ** or ##, $p < 0.01$; *** or ###, $p < 0.001$. RPKM: Reads Per Kilobase of transcript, per Million mapped reads.

---

### Table 1 Plasma triglycerides, cholesterol, and adipokine levels in female (F) and male (M) offspring.

| Maternal diet | moC | | moHF | |
| --- | --- | --- | --- | --- |
| Sex | F | M | F | M |
| Total TG | 0.54 ± 0.04 | 0.58 ± 0.02 | 0.65 ± 0.05 | 1.01 ± 0.16#* |
| Total Chol | 2.7 ± 0.2 | 2.4 ± 0.3 | 3.2 ± 0.2# | 3.6 ± 0.3## |
| VLDL-Chol | 0.07 ± 0.01 | 0.07 ± 0.01 | 0.08 ± 0.01 | 0.09 ± 0.02 |
| LDL-Chol | 0.44 ± 0.04 | 0.28 ± 0.03* | 0.48 ± 0.08 | 0.75 ± 0.23# |
| HDL-Chol | 2.1 ± 0.1 | 1.8 ± 0.1* | 2.7 ± 0.1# | 2.9 ± 0.1### |
| PAI-1 (mM) | 4241 ± 469 | 7390 ± 997* | 4267 ± 387 | 9633 ± 1717* |
| Ghrelin (mM) | 4583 ± 341 | 5165 ± 181 | 5074 ± 698# | 4673 ± 40p=0.05 |
| GIP (mM) | 705 ± 101 | 753 ± 217 | 2716 ± 743# | 764 ± 29* |

Animals were fasted for 2 h prior the blood collection. Data are presented as mean ± sem. F: female; M: male; *, M versus F and #, moC versus moHF. * or #, $P < 0.05$; ##, $P < 0.01$; ###, $P < 0.001$. For F-moC, $n = 5$; for M-moC, $n = 6$; for F-moHF, $n = 6$; for M-moHF, $n = 4$.

---

have overall better metabolic balance than males, possibly due to female-dependent regulation of their hepatic transcriptional activity, which may protect female offspring in utero from the adverse effect of MO later in life.

**Maternal obesity alters endocrine parameters and hepatic triglyceride profile in the offspring in a sex-dependent manner.** Obesity and HFD are factors that provoke changes in circulating lipids and cytokines. While F-moHF had unchanged total triglyceride (TG) compared to F-moC, their male counterparts displayed elevated levels (Table 1). Total and HDL-cholesterol levels were increased in MO offspring of both sexes (Table 1). MO differently adjusted circulating levels of markers of insulin sensitivity and glucose homeostasis (PAI-1, ghrelin, and GIP)[24,25] in females and in males. The PAI-1 level, a strong predictor of type-2 diabetes and metabolic syndrome[26], was higher in males than in females in both groups and, MO induced and reduced Ghrelin and GIP in females and males, respectively. These results showed that MO affected circulating adipokines positively in females but negatively in males.

Obesity and insulin resistance are associated with hepatic lipid disorders, including liver steatosis, which can further develop into hepatocellular carcinoma[27]. Proton magnetic resonance spectroscopy is the commonly applied method to track TG in real time (Fig. 2a). The fraction of lipid mass was unchanged by MO in both sexes but was higher in M-moHF than F-moHF at MID (Fig. 2b). MO induced the fraction of saturated lipids only in females at MID (Fig. 2c), whereas MO severely reduced the fraction of monounsaturated lipids in males at END (Fig. 2d). At MID, MO reduced the fraction of polyunsaturated lipids in females, inversely at END, MO induced the fraction of polyunsaturated lipids in males (Fig. 2e).

Changes in hepatic TG profile have been associated with several metabolic diseases including insulin resistance, metabolic-associated fatty-liver diseases and hepatocellular carcinoma[28,29]. We analyzed in depth the hepatic TG composition at END using lipidomics. Overall, TG classes were sex-dependently changed, with M-moC having proportionally more TG46, TG56, TG58, and TG60 but less TG54 than F-moC (Fig. 2f–h). Interestingly, MO tended to reduce the proportion of short chain TG (TG46, TG48, TG50, and TG51) but increased long-chain TG (TG56, TG58, and TG60) only in females. TG species within each TG class were also highly sex-dependent, MO remodeled TG species mostly in females (Fig. 2i and Fig. S2). We next assessed the degree of TG bonds since the saturation profile of hepatic TG has been correlated to several metabolic dysfunctions. We detected fewer TG-containing 3- and 4-double bonds in males than females regardless of the maternal diet (Fig. 2j). MO reduced the

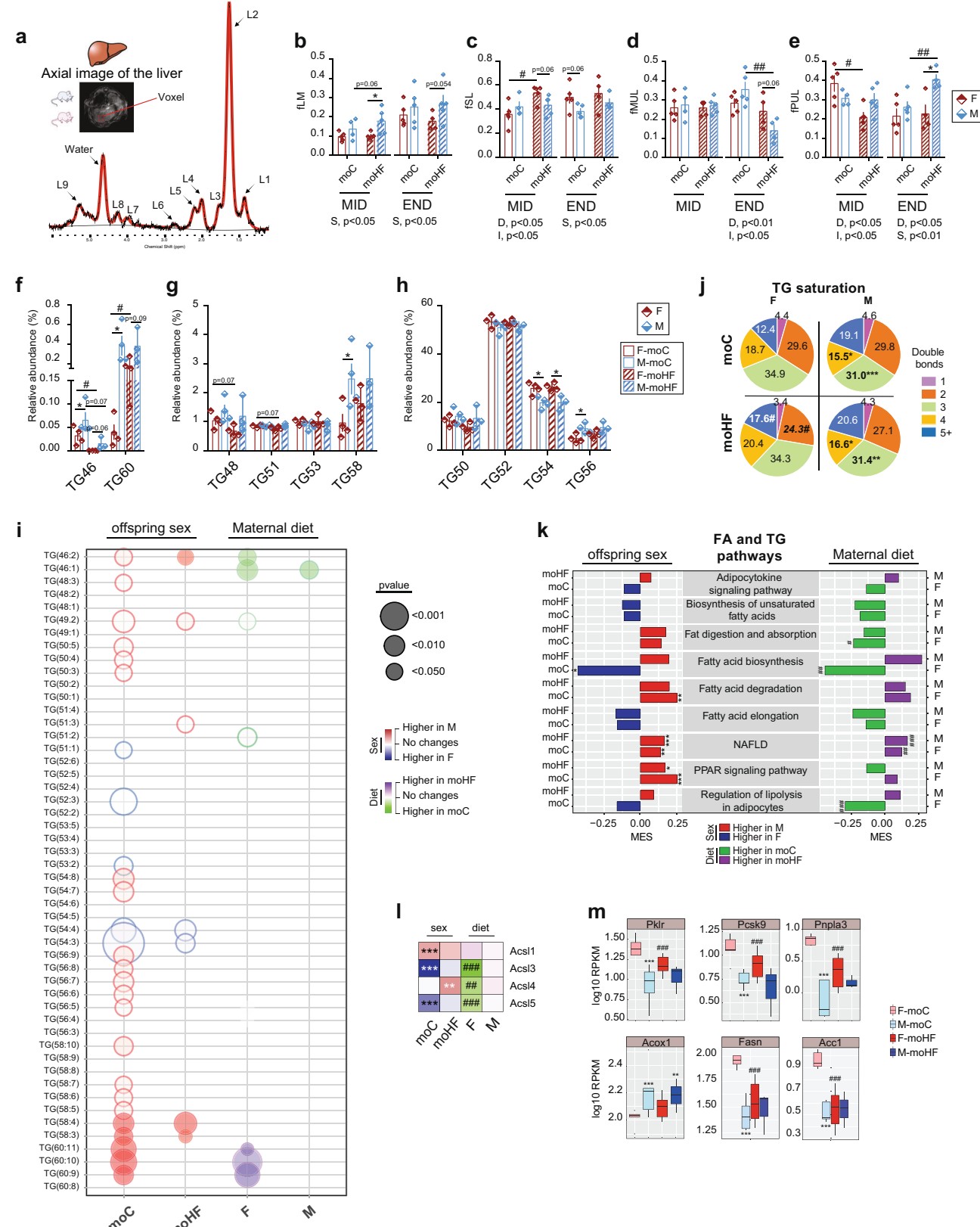

proportion of TG-containing 2-double bonds in females, which is in line with the modulation of the desaturases (*Scd1*, *Fads1/2*) gene expression by MO (Table S2).

Fatty acids as part of TG molecules act as energy source and signaling molecules that can modulate metabolic response in obesity. We found that fatty acids composition was sex-dependent whereby the abundance of the C20:0, C20:1ω9, C20:2ω9, and C20:3ω6 species was higher in males than females, irrespective of the maternal diet. MO increased the abundance of C18:2ω6 and reduced C16:0 species in females (Fig. S3a). Males

**Fig. 2 Maternal obesity adjusts the triglycerides composition in the liver of offspring and causes sex-dependent transcriptional alterations.** Female (F, red bars) and male (M, blue bars) offspring born from C diet mothers (moC, open bars) and from HF diet mothers (moHF, stripped bars) at MID and END. **a** Representative axial image of the liver with single voxel spectroscopy and one representative proton spectrum used for in vivo quantification of the fraction of **b** lipid mass (fLM), **c** saturated lipids (fSL), **d** monounsaturated lipids (fMUL), and **e** polyunsaturated lipids (fPUL). **f–h** Relative abundance of TG classes in the liver; **i** bubble plot of the significantly changed relative level of TG species in liver extracts; **j** pie charts showing the hepatic TG saturation profile; **k** bar plot presenting the MES between sexes in moC and moHF (left panel), and in response to MO in F and M (right panel) of the KEGG pathways involved in the FA and TG metabolism. Red and blue bars indicate higher expression in M and F, respectively, and, green and purple bars indicate higher expression in moC and moHF, respectively. **l** Heatmap of the log2 fold change expression levels of the *Acsl* family genes, and **m** Box plots showing expression (RPKM, log10) of selected genes involved in the FA and TG pathways. For **b–e**, F-moC ($n = 5$), M-moC ($n = 9$), F-moHF ($n = 5$), and M-moHF ($n = 5$). For **f–j**, F-moC ($n = 4$), M-moC ($n = 4$), F-moHF ($n = 4$), and M-moHF ($n = 3$). For **k–m**, F-moC ($n = 5$), M-moC ($n = 5$), F-moHF ($n = 6$), and M-moHF ($n = 3$). For **b–h**, data are presented as mean ± sem; for **m**, data are presented as mean ± sd. For **b–e**, two-way ANOVA (sex (S), mother diet (D), interaction and (I) between sex and diet) followed by Tukey's multiple comparisons test when significant ($p < 0.05$). For **l**, **m**, differences between two groups (sexes, F versus M; maternal diet moC versus moHF) were determined by unpaired two-tailed *t*-test corrected for multiple comparisons using the Holm–Sidak method, with alpha = 5.000%. For pathway and DE genes analysis we used the Benjamini–Hochberg correction with FDR < 0.1, when significant. *, M versus F and #, moHF versus moC, $p < 0.05$; ** or ##, $p < 0.01$; *** or ###, $p < 0.001$.

had globally more of the fatty acids containing 3-double bounds than females, irrespective of the maternal diet, but MO increased polyunsaturated fatty acids in females (Fig.S3b). Desaturation of fatty acids is controlled by desaturase enzymes. The desaturase activity Δ9 was unchanged between groups but Δ5 was higher in females than in males in both groups, supported by the higher expression level of *Fads1/2* genes (Fig. S3c and Table S2).

These physiological parameters were in alignment with the transcriptional profile. We observed that genes controlling metabolic pathways involved in fatty acids breakdown (PPAR signaling and FA degradation) and fatty liver diseases were higher expressed in males than in females in both groups (Fig. 2k, left panel). In contrast, the activity of genes of the fatty acid biosynthesis pathways was higher in F-moC than in M-moC and reduced by MO only in females. MO reduced expression of genes controlling fat digestion and absorption as well as adipocyte lipolysis activity in females, and MO increased expression of gene regulating fatty liver diseases in both sexes (Fig. 2k, right panel). We next inspected genes involved in the selected lipid pathways and found many DE genes between sexes in moC and much less in moHF. These changes were explained by MO remodeling the activity of specific gene in females but not males (Table S2). The expression level of most of the DE genes involved in lipid homeostasis, namely the long-chain acyl-CoA synthase family members (*Acsl1/3/4/5*) (Fig. 2l) as well as *Pklr*, *Pcsk9*, *Pnpla3*, *Acox1*, *Fasn*, and *Acc1* (Fig. 2m) was higher in F-moC than in M-moC and were reduced by MO in females only, except for *Acox1*. *Acox1* expression was unchanged by MO and was higher expressed in males than in females (Fig. 2m). Modulation of these genes affects hepatic intracellular TG levels and the viability of these cells[30–32]. These sex differences possibly appeared through differences in *Esr1* and *Ar* gene expression (Fig. S1b), both critical regulator of hepatic lipid metabolism[23,33].

In sum, hepatic fatty acids and TG composition is sex-dependent, and MO remodeled their profiles differently in female and male offspring, which lead to sex-dependent liver dysfunctions later in life. Females born from obese mothers have more polyunsaturated TG than males and reduced TG-containing 2-double bounds compared to F-moC. Moreover, MO modulates the hepatic transcriptional activity in females only, which may be a key contributor to the sexual dimorphism in obesity-associated liver disorder in adulthood.

**Sex-dependent hepatic phospholipid profiles are altered in the offspring born from obese mothers.** In hepatic cells, phospholipids comprise the most abundant lipid class. Phospholipids are found in the plasma membrane and intracellular organelles, and

the lipidome of each organelle may be remodeled by extra- and intra-cellular stimulations that may also affect lipid trafficking across the membrane and the organelles. We comprehensively profiled hepatic phospholipids using a LC-MS lipidomic approach. Principal component analysis separated the phospholipids classes into two distinct groups clustered by sexes (Fig. 3a), which indicates that phospholipids profile is strongly sex-dependent. We found four major subclasses of phospholipids, phosphatidylcholine (PC), lysoPC (LPC), phosphatidylethanolamine (PE), and lysoPE (LPE). The relative abundance of PC and LPC was similar between sexes, but males overall tended to have higher levels of PE than females and LPE abundance was higher in males than in females regardless of the maternal diet (Fig. 3b–g and Fig. S4a, S4b). Abundances in the PC and PE classes differed between sexes in moC group affecting 15/30 PC and 8/25 PE species. MO reduced these differences considerably, to 5/30 for PC and 2/25 for PE. This was largely the result of remodeling PC and PE species in females (Fig. S4c, S4d). Similarly, the saturation profiles of PC and PE species were sex-dependent in moC-offspring, and MO abolished these differences (Fig. 3e). The saturation profile was highly variable between sexes in moC but not in moHF group (Fig. 3h, i), in line with the overall remodeling of metabolic responses in female offspring by MO.

The low abundant subclasses of phospholipids, phosphatidylserine, phosphatidylglycerol, cardiolipin and phosphatidylinositol and two sphingolipids, ceramides and sphingomyelin were detected by LC-MS (Fig. 3b). Although no sex differences were observed within the phosphatidylserine class, MO reduced the relative levels in 3/7 and 5/7 PS species in females and males, respectively (Fig. 3j and Figs. S5a, S5b). Moreover, the phosphatidylserine saturation profile was affected by MO and was sex-dependent in moHF (Fig. 3k). Phosphatidylglycerol class and species were similar between sexes (Fig. 3b). Cardiolipin class and species were more abundant in females than males, especially in the moC group. MO tended to reduce cardiolipin level in females with no differences in the saturation profile between all groups (Fig. 3l, m and Fig. S5c). Phosphatidylinositol classes and species were more abundant in males than in females regardless of the maternal diet, with males having more of the phosphatidylinositol-containing 3- and 2-double bonds than females in moC and moHF respectively. F-moC showed more of the phosphatidylinositol-containing +4-double bonds than M-moC (Fig. 3n, o and Fig. S5d). The Ceramides class was induced by MO in both sexes, with females having more of the Cer(d34) and Cer(d36) and less Cer(d40) classes than males in both groups (Fig. 3p and Fig. S5e). Of note, the glucosylceramide species, which affects the function of cells, and may in term lead to liver diseases[34], were reduced and induced by MO in females

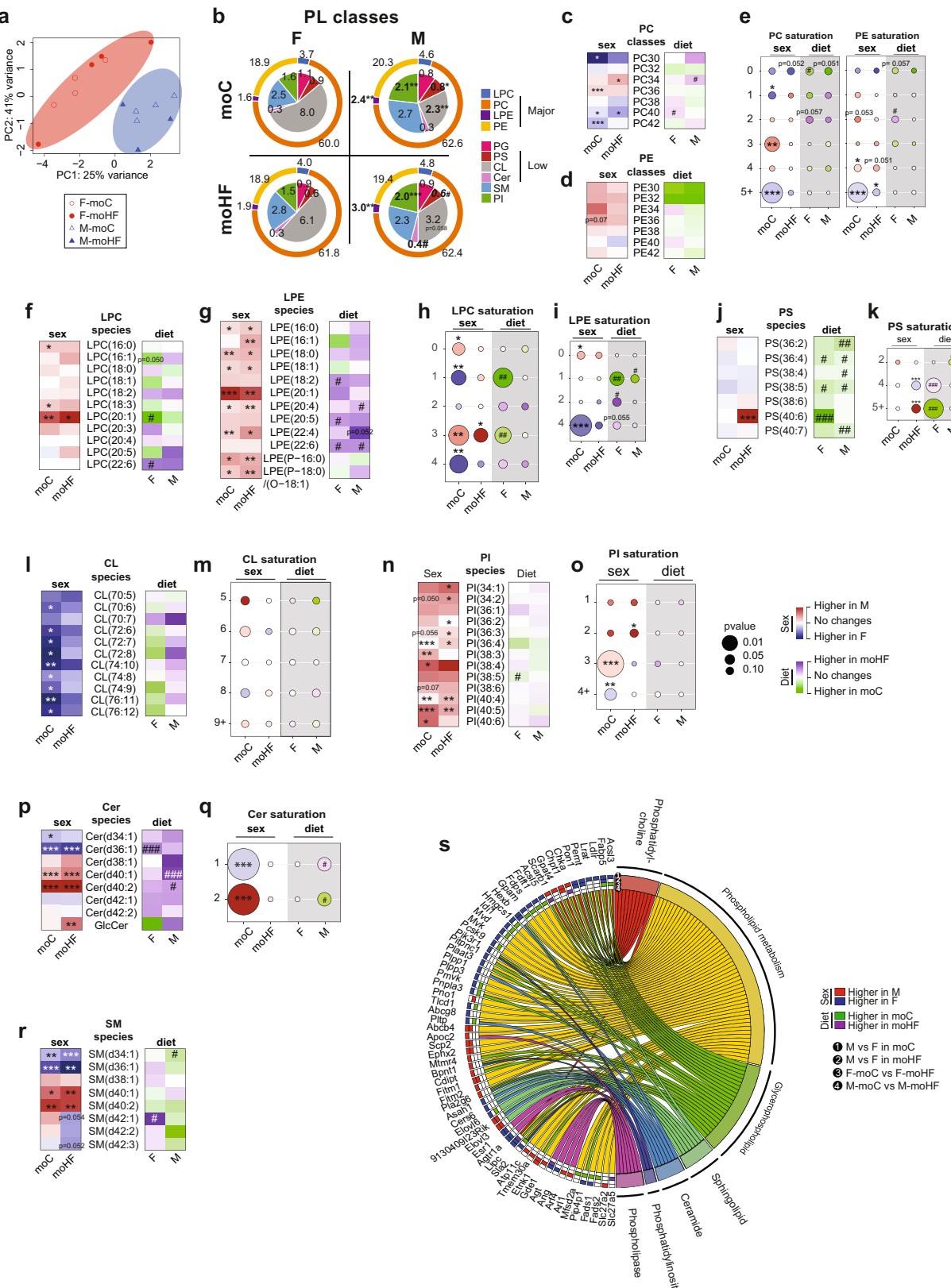

and males respectively, leading to higher abundance in males. F-moC had more of the ceramides containing 1-double bond and less of those containing 2-double bonds than M-moC (Fig. 3q). Females' and males' livers had similar sphingomyelin class abundance but contained different sphingomyelin species in both groups (Fig. 3r and Fig. S5f). Our gene ontology enrichment

analysis established that sex is a major transcriptional regulator of the phospholipids pathways and indicated that MO remodeled gene expression mainly in females (Fig. 3s).

In conclusion, phospholipids classes and species are highly sex-dependent. Overall, females have more cardiolipin which is essential for mitochondria function, and less pro-inflammatory

**Fig. 3 Offspring hepatic phospholipids composition is sex-dependent. a** Principal component analysis (PCA) plot of the phospholipid profile in offspring livers. **b** Chart representing relative abundances of all PL classes categorized in low (inner layer of the chart) and major classes (outer part of the chart); Heatmaps presenting the log10 fold change between sexes (red and blue boxes) in moC and moHF and in response to MO (green and purple boxes) in F and M of the **c** phosphatidylcholine (PC) and **d** phosphatidylethanomine (PE) lipid classes. **e** Bubble charts showing the log10 fold change of the saturation profile in PC and PE classes between sexes (moC and moHF columns, white background) and in response to MO (F and M columns, gray background); Heatmap presenting the log10 fold change between sexes in moC and moHF and in response to MO in F and M of the **f** lysoPC (LPC), **g** lysoPE (LPE), **j** phosphatidylserine (PS), **l** cardiolipin (CL), **n** phosphatidylinositol (PI), **p** ceramide (Cer), and **r** sphingomyelin (SM) lipid species; bubble charts showing the log10 fold change of the saturation profile in **h** LPC, **i** LPE, **k** PS, **m** CL, **o** PI, and **q** Cer classes between sexes (moC and moHF columns, white background) and in response to MO (F and M columns, gray background). **s** Chord graph presenting the genes associated with phospholipid pathways. The significant expression of each gene based on log2 fold change is presented as (i) red boxes for upregulation in M, (ii) blue boxes for upregulation in F, (iii) green boxes for upregulation in moC, (iv) purple boxes for upregulation in moHF, and (v) white boxes when not significant. For **a–r**, F-moC ($n = 4$), M-moC ($n = 4$), F-moHF ($n = 4$), and M-moHF ($n = 3$). For **s**, F-moC ($n = 5$), M-moC ($n = 5$), F-moHF ($n = 6$), and M-moHF ($n = 3$). For gene expression analysis (**s**) we used the Benjamini–Hochberg correction with FDR < 0.1, when significant. Differences between two groups (sexes, F versus M; maternal diet moC versus moHF) were determined by unpaired two-tailed $t$-test corrected for multiple comparisons using the Holm–Sidak method, with alpha = 5.000%. *, M versus F and #, moHF versus moC, $p < 0.05$; ** or ##, $p < 0.01$; *** or ###, $p < 0.001$.

mediators such as lysophospholipids, phosphatidylinositol, glucosylceramides, and sphingolipids than males which is correlated with balanced metabolic profile[34,35]. Moreover, cardiolipin is the signature lipid of mitochondria, which would indicate more mitochondria mass in females' livers and hence different capacity for oxidative phosphorylation[36] which will benefit the metabolic profile.

**Transcriptional and posttranscriptional regulation of metabolic pathways in offspring liver is sex- and maternal diet-dependent.** In accordance with the magnetic resonance spectroscopy data, hematoxylin-eosin-stained liver sections also revealed that males had higher number and lager lipid droplets compared to females (Fig. 4a). These lipid droplets were likely formed by TG and phospholipids that are central in controlling fatty liver diseases[28,37]. Moreover, dysfunctional TG and phospholipids can initiate endoplasmic reticulum stress and inflammation[38]. To understand the associated transcriptional changes in offspring's liver, we inspected the DE genes. We found that the number of DE genes by MO was higher in females than males (325 versus 33, respectively). Only four DE genes were shared between sexes (Fig. 4b). When compared to females, males showed higher expression levels of *Sult2a8* and *Cyp2c54*, a lower expression level of *Cyp2c37* and same level of *Cyp2c50* in both groups (Fig. 4c). DE gene between sexes in moC and moHF (Fig. 4d) revealed that nearly half of genes (46%) were exclusively differentially expressed in the moC group and about one third (32%) only in the moHF group. Only 22% of the DE genes were shared between moC and moHF conditions (Fig. 4d).

In mammals, including humans, the pair of sex chromosomes determines the biological sex. A plethora of characteristics can be determined by sex chromosomes and more importantly by X-linked chromosomes. Therefore, we inspected whether genes encoded by the sex chromosomes were affected by the maternal diet. We found that MO resulted in six and one relevant to metabolism DE genes in females and males, respectively, all linked to the X-chromosome (Fig. 4e, f). In females, MO reduced the expression of *Nsdhl* and *Acls4* genes, which are involved in lipid and cholesterol metabolism[39,40]. Interestingly, MO altered expression of *G6pdx*, *Rbm3*, *Ctps2*, and *Apex2* genes, that are linked to hepatocarcinoma development[41–44]. In males, MO reduced expression of *Zdhhc9* gene, involved in cancer and metabolism[45] (Fig. 4f, g). Importantly, expression of X-linked genes *Apex2*, *Acls4*, and *Pgrmc1*, involved in lipid metabolism and hepatocarcinoma was sex-dependent[40,41,46]. The Y-chromosome encodes genes *Uty*, *Kdm5d*, *Ddx3y*, and *Eif2s3y* were higher expressed in M-moC than in F-moC. Upon moHF, the expression

of X-linked genes was reduced in both sexes. Expression of *Il13ra1* and *Kdm5c* encoding for liver glucose[47] and lipid[48] homeostasis was higher in females, and expression of *Atp11c* and *Acsl4* encoding for genes connected to lipid disorders and hepatocarcinoma development[40,49] was higher in males.

We then performed a GO enrichment analysis of the DE genes and extracted the top 10 enriched GO terms for up- and downregulated genes to identify biological processes, molecular functions, or cellular components affected by sex and/or maternal diet. In females, MO upregulated metabolic and catabolic processes and downregulated immune and inflammatory processes. In males, MO upregulated xenobiotic and fatty acid metabolic processes and downregulated biosynthetic processes, while regulating a low number of genes (Fig. 4i). When comparing both sexes in the moC group, males had higher catabolic processes and lipid metabolism pathway activity and lower lipid and steroid biosynthetic activity than females (Fig. 4j). Upon moHF, males showed higher metabolic and catabolic processes and a lower immune and inflammatory pathways activity than females.

Altogether these results indicated that MO alters the liver transcriptome to a much larger extent in females than in males. We confirmed that MO reprograms the transcription of genes in offspring liver in a sex-dependent manner, which defined metabolic response. Several DE genes were X-linked and could contribute to the sexual dimorphism.

Given that the content and the composition of hepatic lipids were sex-specifically regulated towards higher lysophospholipids, phosphatidylinositol, and glucosylceramides content in males than in females and that hepatic lipids mediate inflammation, we explored pathways involved in inflammation (Fig. 4k). Our analysis unveiled that the activity of genes controlling inflammatory pathways was higher in F-moC than in M-moC. These differences were abolished by MO, whereby MO induced inflammatory pathway activity mainly in males. In contrast, females induced gene expression related to apoptosis. These results imply a sex-dependent regulation of inflammatory pathways in offspring' liver, and that MO modulates these pathways differently in both sexes. Within the inflammatory pathways, we found that inflammatory gene expression is sex-dependent irrespective of the maternal diet, and MO altered the expression of a few genes only in females (Table S3). Among those, we found genes belonging to the cathepsin (*Cts*) family that drive liver inflammation and fibrosis[50] (Fig. 4l). Interestingly, the expression level of all *Cts* genes was higher in F-moC than in M-moC but MO reduced the expression level of *CtsD* genes in females. In sum, inflammatory gene expression in liver is highly sex-

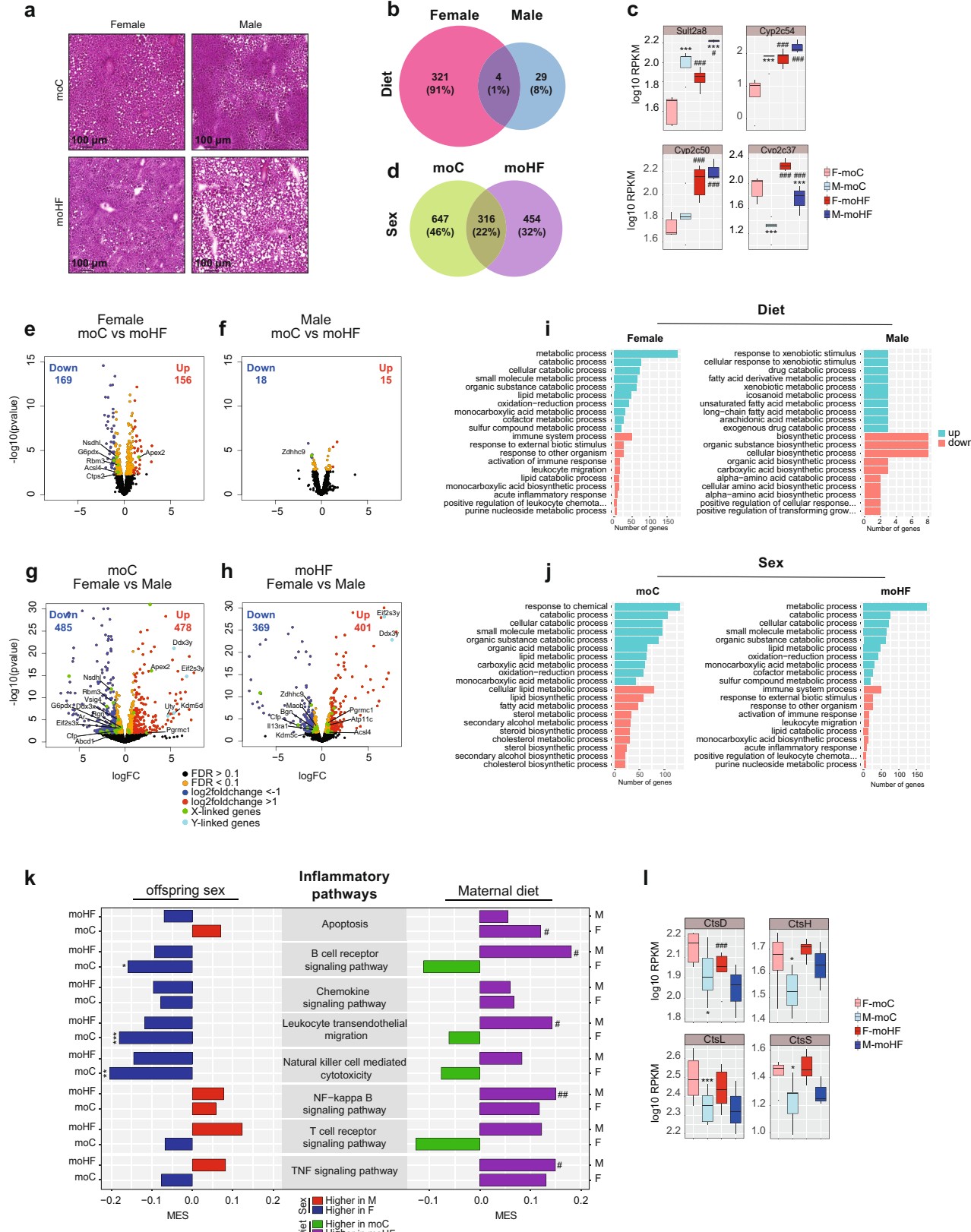

dependent and MO appears to prevent liver injury in females by reducing inflammatory processes.

**Maternal obesity may prevent hepatocarcinoma development in female offspring.** Recent research in animal models has elucidated potential programming mechanisms that alter hepatic

function[51–54] and cellular signaling responses[55,56]. Our histological analysis of offspring's liver revealed the presence of large regions within the liver exhibiting high cell proliferation in females, in contrast to small and scattered proliferative spots in males. Importantly, MO reduced these areas of cell proliferation in females, which is in line with reduced expression levels of

**Fig. 4 Maternal obesity promotes hepatic inflammatory response in male offspring. a** Hematoxylin-eosin (H&E) staining of frozen liver sections from moC and moHF offspring; Venn diagram of the DE genes in response to MO, in **b** females (F) and males (M) and between sexes in **d** moC and moHF. **c** Box plots showing expression (RPKM, log10) of the four DE genes by MO shared between sexes; Volcano plots of the DE genes in response to MO in **e** F and **f** M and between sexes in **g** moC and **h** moHF with selected genes linked to the X- and Y-chromosomes. Genes residing on the X-chromosome are marked in green and those on the Y-chromosome are marked in light blue; top 10 significantly up and top 10 significantly down enriched biological GO terms in response to MO in **i** F and M; and between sexes in **j** moC and moHF. **k** Bar plot presenting the MES between sexes in moC and moHF (left panel) and in response to MO in F and M (right panel) of the KEGG pathways involved in inflammation. Red and blue bars indicate higher expression in M and F, respectively, and green and purple bars indicate higher expression in moC and moHF, respectively. **l** Box plots showing expression (RPKM, log10) of selected genes involved in the inflammatory pathways. For volcano plots: Significantly upregulated (log2FC > 1) and downregulated (Log2FC < −1) genes are presented as red and blue dots, respectively. Orange dots indicate the genes that are significantly changed (FDR < 0.1). Black dots indicate not significant (FDR > 0.1). For **a** F-moC ($n = 3$), M-moC ($n = 3$), F-moHF ($n = 4$), and M-moHF ($n = 2$). For **b**–**l**, F-moC ($n = 5$), M-moC ($n = 5$), F-moHF ($n = 6$), and M-moHF ($n = 3$). For **c** and **l**, data are presented as mean ± sd. Differences between two groups (sexes, F versus M; maternal diet moC versus moHF) were determined by unpaired two-tailed $t$-test corrected for multiple comparisons using the Holm–Sidak method, with alpha = 5.000%. For **i**, **j**, significance was determined using Fisher's Exact test with $P$ values ≤0.05. For pathway and DE genes analysis we used the Benjamini–Hochberg correction with FDR < 0.1 when significant. *, M versus F and #, moHF *versus* moC, $p < 0.05$; ** or ##, $p < 0.01$; *** or ###, $p < 0.001$.

*Rbm3*, *Ctps2*, and *Acsl4* X-linked genes[40,42,43] (Figs. 4e and 5a). By exploring KEGG pathways associated with cancer, we found that the expression of these pathways was higher in females than in males, regardless of the maternal diet, but MO repressed the activity of the pathways involved in cell cycle control and resulted in higher activity of chemical carcinogenesis, notch signaling and retinol metabolism pathways in females compared to males. In contrast, MO reduced the expression of genes controlling notch signaling pathway activity and induced retinol metabolism in males (Fig. 5b). We inspected DE genes controlling selected cancer pathways (Table S4) and identified genes belonging to large gene families, namely UDP-glycosyltransferases (*Ugt*) and the sulfotransferases (*Sult*), which were highly sex-specifically expressed. For example, *Ugt3* and *Ugt2* genes were higher expressed in males than in females, whereas the expression of *Ugt1* and nearly all *Sult* genes, except for *Sult2a8*, were higher expressed in females (Fig. 5c, d). MO increased the expression level of the *Ugt* genes in both sexes (Fig. 5c). Remarkably, we found two key genes (*Osgin1* and *Stat1*) that are known tumor repressors[57,58]. *Osgin1* expression was higher in M-moC than F-moC, and both *Osgin1* and *Stat1* were overexpressed in females by MO (Fig. 5e). In line with this, genes promoting cancer development and cell apoptosis (*Ccnd1*, *Fdps*, and *Pik3r1*) were lower expressed in M-moC than in F-moC, and MO reduced the expression in females only (Fig. 5f). Overall, MO may reduce cell proliferation by inducing the expression of tumor suppressor and cell apoptosis genes in females, which would result in protection from MO in the female offspring.

Collectively our data demonstrate that MO modulates the metabolism differently in female and male offspring by reprogramming hepatic transcriptional activity in a sex-dependent manner. We showed that sex and MO alter the regulation of genes, often encoded on sex chromosomes, that control major metabolic processes in the liver of the offspring which could contribute to the observed sexual dimorphism in response to maternal obesity-associated metabolic risks. Based on our results, we postulated mechanisms through which MO balanced female but not male offspring from metabolic impairment when exposed to HFD in utero and after weaning (Fig. 6). In livers of female offspring a reduced expression of lipogenesis and pro-inflammatory genes resulted in remodeling of TG species and decreased risks of hepatocarcinoma. However, a slight reduction of hepatic insulin sensitivity was observed. These effects were associated with an increased activity of oxidative phosphorylation and browning pathways in adipose tissue[15] (Fig. 6a). In contrast, livers of male offspring showed severe hepatic steatosis and increased inflammation, possibly due to high hepatic levels of phosphatidylinositol, lysophosphatidylethanolamine and

glucosylceramides and negative feedback mechanisms from the adipose tissue[15] (Fig. 6b).

## Discussion

This study highlighted mechanistic differences through which the maternal diet can prime regulation of hepatic metabolism in female and male offspring. We previously demonstrated that MO leads to a sexually dimorphic reprogramming of hepatic lipid composition and gene expression, and that this also occurs when the offspring received a control diet after weaning[13]. We also showed that other organs are affected in a sex-dependent manner[15]. Here, we showed that MO offspring fed an obesogenic diet have sex-specifically altered liver lipidomes explained by accompanying changes in the transcriptome. When compared to males born from lean mothers, males from obese mothers showed higher visceral on total fat ratio and glucose intolerance at early life stage. M-moHF had lower body weights early after birth (3–9 weeks old), while their growth accelerated spontaneously at a later stage (10–15 weeks old) which is known to be associated with insulin resistance later in life[59]. At MID, M-moC had similar glucose tolerance to F-moC but they displayed lower insulin sensitivity, which is a reasonable approximation of direct measure of insulin resistance in both rodents[60] and humans[61]. At END, males showed impaired glucose tolerance and insulin sensitivity compared to females, together with increased type II diabetes pathway activity and higher circulating level of PAI-1. On the contrary, F-moHF showed increased circulating levels of ghrelin and GIP compared to F-moC and to M-moHF, which correlates to insulin sensitivity[24,25], associated with a reduction of type II diabetes pathway activity. Three genes known to act as major regulators of insulin sensitivity[18–20], 3-phosphoinositide-dependent protein kinase-1 (*Pdk1*), NAD(P)H oxidase 4 (*Nox4*), and prolactin receptor (*Prlr*), were influenced by the sex and are controlled by estrogen[62,63]. These differences in hormonal secretion and transcriptional activity could be inherited from in utero environment[17,64,65]. Results from a recent study are in line with our current findings in liver and highlight the existence of sexually dimorphic responses to MO in the fetal heart and identify lipid species that might mediate programming of metabolic health[65]. Most importantly, Pantaleão et al.[65] showed that female fetal heart lipidomes are more sensitive to MO than males, as we found in the liver in our current study. It is of interest to note, that males from obese mothers feed with a control diet after weaning, show improved insulin sensitivity at END[13]. This indicates that the changes induced by MO in utero in male offspring can be reversed by a post-weaning diet.

By using cross-sectional data analysis obtained by multidisciplinary techniques, we showed that lipid profile in the liver of

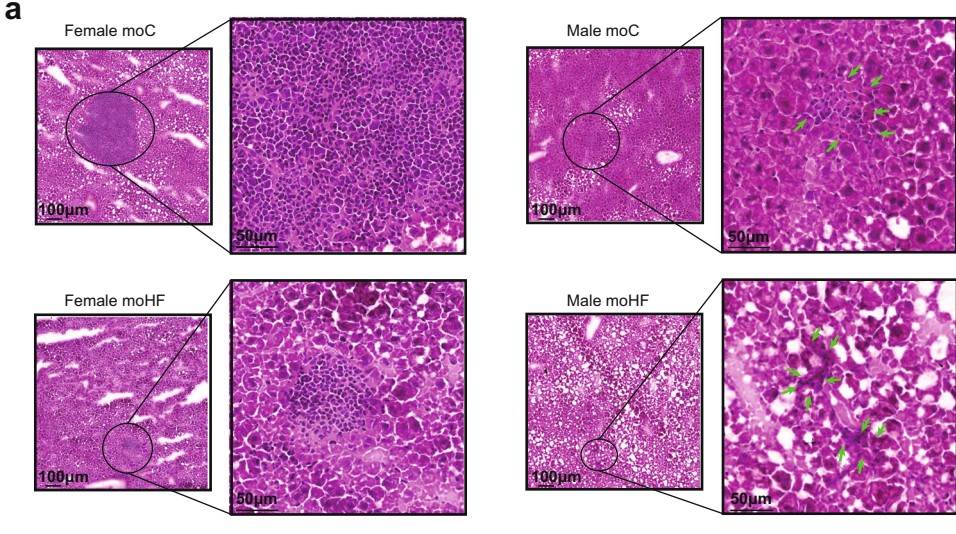

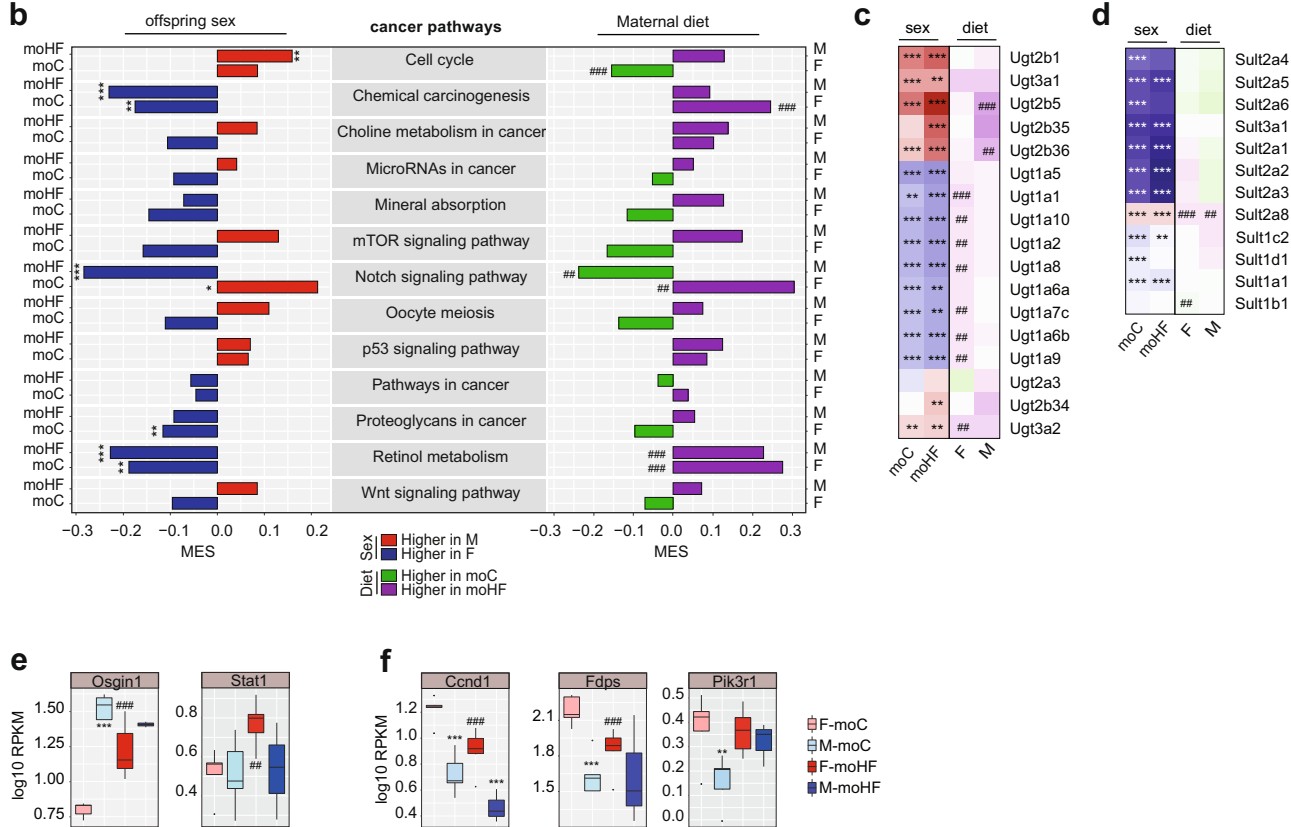

**Fig. 5 Maternal obesity has sex-dependent effects on hepatocarcinoma progression in obese offspring. a** 10x and 40x hematoxylin-eosin (H&E) images from liver sections of F-moC, M-moC, F-moHF, and M-moHF. **b** Bar plot presenting the MES between sexes in moC and moHF (left panel) and in response to MO in F and M (right panel) of the KEGG pathways involved in hepatocellular carcinoma. Red and blue bars indicate higher expression in M and F respectively and, green and purple bars indicate higher expression in moC and moHF, respectively. Heatmap of the log2 fold change expression levels of the **c** *Ugt* - gene family and **d** *Sult* - gene family; **e, f** Box plots showing expression (RPKM, log10) of genes involved in the nominated cancer pathways. For **a**, F-moC ($n = 3$), M-moC ($n = 3$), F-momHF ($n = 4$) and M-moHF ($n = 2$). For **b–f**, F-moC ($n = 5$), M-moC ($n = 5$), F-moHF ($n = 6$), and M-moHF ($n = 3$). For **e, f**, data are presented as mean ± sd. Differences between two groups (sexes, F versus M; maternal diet moC versus moHF) were determined by unpaired two-tailed *t*-test corrected for multiple comparisons using the Holm–Sidak method, with alpha = 5.000%. For pathway and DE genes analysis we used the Benjamini–Hochberg correction with FDR < 0.1, when significant. *, M versus F and #, moHF versus moC, $p < 0.05$; ** or ##, $p < 0.01$; *** or ###, $p < 0.001$.

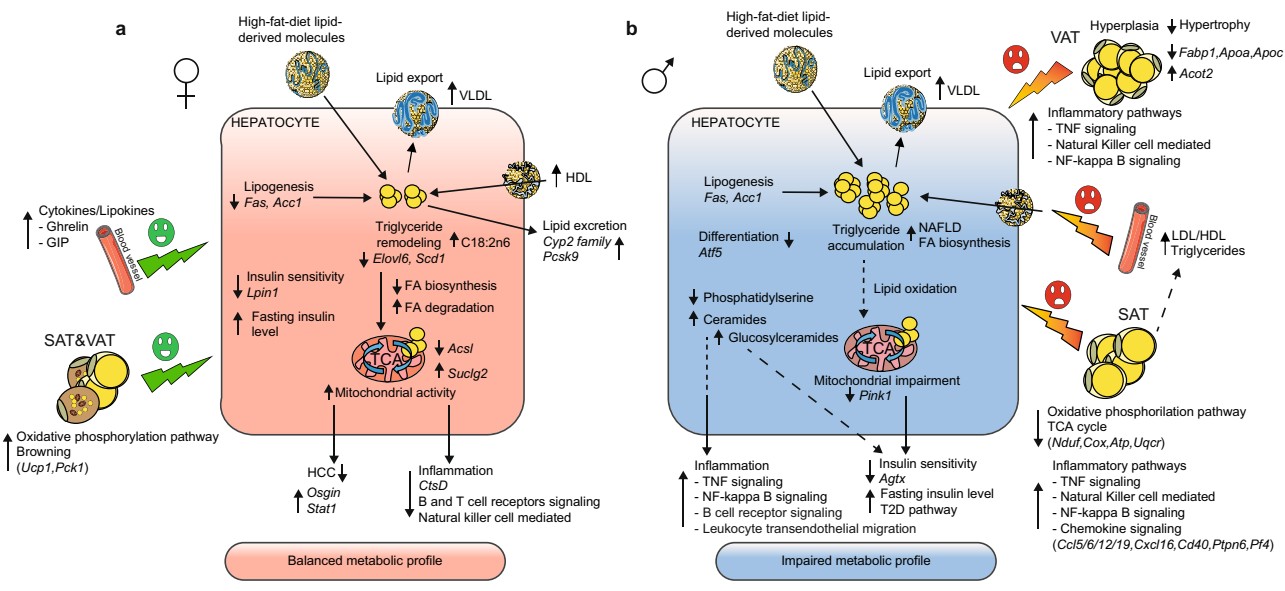

**Fig. 6 Sexually programmed metabolic response in the obese offspring exposed to maternal obesity. a** Livers from female offspring demonstrate reduced insulin sensitivity and decreased lipogenesis and inflammatory pathways, decreased hepatocarcinoma and remodeling of triglyceride species. Oxidative phosphorylation and browning processes are increased in SAT and VAT, respectively. Circulating cytokine and lipokine levels, ghrelin and GIP are higher in F-moHF compared to F-moC and to M-moHF. Altogether, these metabolic adaptations would protect the liver from later metabolic complications in response to MO. **b** Livers from male offspring demonstrate induced lipogenesis and steatosis, induced glucosylceramides production and increased inflammation possibly due to negative feedback signaling from the adipose tissue (reduction of oxidative phosphorylation associated with an induction of inflammatory pathways) which would promote metabolic disorders. Arrows stand for up-↑ and down-↓ regulation. HCC, hepatocellular carcinoma; SAT, subcutaneous adipose tissue; VAT, visceral adipose tissue; VLDL, very low; LDL, low and HDL, high-density lipoproteins.

the offspring was distinct for each sex and the differences were associated with sex-dependent modulation of transcriptional activities. In previous studies, we and others uncovered that relative abundancy of hepatic phospholipids, TG and fatty acids species were different between sexes[16,66,67], which may be a key element in the sex-specific metabolic complications in obesity. Here, we showed that male offspring from obese mothers contained more toxic hepatic phospholipids (lysophospholipids, phosphatidylinositol and glucosylceramides) than females. The sexual dimorphism in the phospholipids profile of MO offspring's liver may differently modulate transmission of biological signals across the cell and lipid droplet membranes between sexes and could be attributed to sex-dependent transcriptional activity of major genes involved in the phospholipid synthesis pathways. We confirmed our previous findings that MO modulates hepatic TG molecular species in female but not in male offspring[13], as observed in fetus heart in a recent study[65]. Desaturases are enzymes that control the balance between saturated, mono- and polyunsaturated FA, which become incorporated into TG and phospholipids. When fed a control diet after weaning, offspring born from obese mothers showed sex-dependent Δ9 desaturase activity but similar Δ5 desaturase activity[13]. In this study, male offspring showed lower Δ5 desaturase activity compared to females in both mother diet groups. Low Δ5 desaturase activity is correlated to insulin resistance, abdominal adiposity and predicts the development of type-2 diabetes[68]. Long chain acyl co-A synthetases (*Acsl*) are important regulators of fatty acids uptake. *Acsl1* promotes TG accumulation in the liver as opposed to *Acsl3* (localized in lipid droplets) and *Acsl5* (in mitochondria), which regulate lipogenesis and β-oxidation, respectively, thereby being essential for lipid homeostasis[30,69]. Importantly, the expression of *Acsl* is controlled by estrogen in mammals[70] and was found to be highly sex- and maternal-diet dependent.

Alterations of hepatic lipid composition can cause liver damage through various processes, including inflammation, oxidative stress, fibrosis, and hepatocellular carcinoma. Expression of cathepsin (*Cts*) genes was higher in females than in males born to lean mothers but reduced by MO in females only. *CtsD* has been identified as a marker of liver inflammation and fibrosis in murine steatohepatitis[50]. In line with this, F-moHF showed smaller lipid droplets and reduced cell proliferation and inflammation as compared to F-moC. UDP-glucuronosyltransferase (UGT) and sulfotransferase (SULT) gene family's expression was sex-specific and maternal diet-dependent. They are essential regulators of xenobiotic and endobiotic metabolism and may be crucial regulators of hepatic cholesterol and lipid homeostasis[41,71]. The mechanism(s) by which in utero MO may protect female offspring from liver dysfunction need to be addressed in more extensive future studies.

The modulation of pathophysiology between sexes relies on the effect of three main classes of sex-biased modulators: sex chromosome (XX versus XY) as well as the long-lasting and transient activation levels of sex hormones during life span[72]. In line with a recent study that explore the role of the sex hormone level, the gonadal sex, and the sex chromosome on gene expression[73], we found that the sex-dependent regulation of metabolism relies on multiple factors, including sex, cell type, and in utero environment. Tissue-specific contributions of major sex-biased factors have been linked to specific transcriptional responses that contribute to systemic metabolic and immune functions. Interestingly, these results point out the complexity of metabolic processes and define a major role of testosterone in liver and of estradiol in adipose tissue. In addition, the relative contribution of each sex-biasing factor to gene regulation were distinct for each tissue type[73]. In our studies, MO led to major changes in gene expression in liver, subcutaneous and brown adipose tissue in female and male offspring[15], highlighting the underappreciated role of sex in the development of metabolic diseases. Further studies would be needed to unravel how the in utero environment results in long-lasting differences in metabolic responses in

female and male offspring. MO did not affect the body weight of obese female offspring, but provoked redistribution of the subcutaneous adipose tissue toward more peripheral and less abdominal accumulation. We recently demonstrated that MO drives transcriptional activities in white adipose toward inflammation in male, and towards browning and oxidative phosphorylation in female offspring[15]. The mechanisms by which MO can differently modulate epigenetic marks in utero between sexes and between tissues remain intriguing and require further investigation.

In conclusion, our detailed studies in mice clearly demonstrate that MO is a preponderant factor for metabolic alterations in offspring. Notably, we show that MO affects hepatic lipid metabolism differently in obese female and male offspring together with sex-specific alterations of the expression of genes involved in insulin signaling, liver steatosis, inflammation, fibrosis, and carcinoma. MO modulates gene expression between sexes as well as between tissues in a sex- and tissue-dependent manner. Interestingly, while MO has negative metabolic consequences in male offspring, it protects female offspring from the development of hepatic steatosis (Fig. 6). However, females from obese mothers showed impaired insulin sensitivity. The identification of several sex- and maternal diet-regulated genes involved in these processes will permit future endeavors of their use to target cardiometabolic risk in humans.

## Methods

**Mice and diet**. All animal procedures were approved by the local Ethical Committee of the Swedish National Board of Animal Experiments. Virgin C57Bl6/J female dams and male sires were received at 4 weeks of age. F0 dams were housed in pairs in six different cages and fed either the control diet (C; D12450H, Research Diets, NJ, USA; 10% kcal fat from soybean oil and lard; $n = 6$, F0-C) or the high-fat diet (HFD; D12451, Research Diets, NJ, USA; 45% kcal fat from soybean oil and lard; $n = 6$, F0-HF) for 6 weeks before mating. Sires remained on control diet until sacrifice. After 6 weeks of their respective diet two F0 dams were mated with one F0 sire. During this short mating period (up to 5 days) sires were on the same HFD as dams in the group (experimental unit). The sires spermatozoa were unlikely affected by the HFD given a general sperm maturation time of ~35 days[74]. After mating, F0 males and pregnant dams were separated. F0 dams were continuously exposed to their respective diets throughout pregnancy and until the end of the lactation period. The F1 offspring were weaned at 3 weeks of age. Afterward, F1 males and females were sex-separated, three to five animals were housed per cage and fed with HFD until the end of the study (Fig. 1a). To simplify the naming convention, the group of offspring born from HFD-fed dams were named "moHF", and the group of offspring born from control diet-fed dams were named "moC". All mice were housed in a 23 °C temperature-controlled 12 h light/dark room, with free access to water and food unless specified. Body weight was recorded weekly throughout the study in all groups. Average food intake in offspring was recorded twice a week for three weeks in four different cages containing grouped mice ($n = 3$–5 animals per cage) at around 4-month of age and at least one week after recovering of in vivo experiments. We then calculated the average food intake per cage during the three experimental weeks. We reported it to the average food intake per mouse according to the number of animals in the cage.

**In vivo magnetic resonance imaging**. Animals were anesthetized using isoflurane (4% for sleep induction and ~2% for sleep maintenance) in a 3:7 mixture of oxygen and air, before being positioned prone in the MR-compatible animal holder. Respiration was monitored during scanning (SA-instruments, Stony Brook, NY, USA). Core body temperature was maintained at 37 °C during scanning using a warm air system (SA-Instruments, Stony Brook, NY, USA). Magnetic resonance imaging images ($n = 5$–7 per group) were collected using a 9.4 T horizontal bore magnet (Varian, Yarnton, UK) equipped with a 40 mm millipede coil, as detailed[5]. Fiji software (http://fiji.sc) was used to compute the volume of fat in different regions of interest in the body. Visceral fat was calculated as the difference between the total fat and the subcutaneous fat signal in the abdominal region. Total subcutaneous fat was calculated as the differences between total fat and visceral fat. Experiment was performed on the same mouse (F1) at the age of 3 months (MID) and 6 months (END).

**In vivo localized proton magnetic resonance spectra**. As for the MRI scanning, animals were anesthetized using isoflurane, respiration was monitored, and core body temperature maintained at 37 °C during scanning. In addition, heart beats were recorded using an electrocardiogram system. Localized proton magnetic resonance spectra ($^1$H-MRS) from the liver ($n = 5$–7 per group) were acquired

from a $2 \times 2 \times 2$ mm$^3$ voxel localized in the left lobe with excitation synchronized to the first R-wave within the expiration period, as detailed[75,76]. Spectroscopy data were processed using the LCModel analysis software (http://s-provencher.com/pub/LCModel/manual/manual.pdf). "Liver 9" was used as a base with all signals occurring in the spectral range of 0 to 7 ppm (water resonance at 4.7 ppm) simulated in LCModel. All concentrations were derived from the area of the resonance peaks of the individual metabolites. Only the fitting results with an estimated standard deviation of less than 20% were further analyzed. $^1$H-MRS spectra revealed nine lipid signals (peaks) in the mouse liver. Peak assignments were based on published data[75,76]. As for the MRI, $^1$H-MRS experiments were repeated twice on the same animal at MID and END.

**In vivo metabolic tolerance tests**. At MID and END, F1 mice were fasted for 6 h prior to the oral glucose tolerance test (OGTT) and for 4 h prior to the insulin tolerance test (ITT), both performed as detailed[77]. Briefly, at time zero (T0) peripheral glucose level was measured at the tail using a One-Touch ultra-glucometer (AccuChek Sensor, Roche Diagnostics) and at T15, T30, T60, and T120 min. For the OGTT, extra blood was collected at each time-point and later plasma was separated by centrifugation (15 min at 2000 RPM) and stored at −80 °C for insulin measurement using a Rat/Mouse Insulin Elisa kit (EMD Millipore - EZRMI13K). The quantitative insulin-sensitivity check index (QUICKI) was calculated using the formula QUICKI = 1/(log(insulinT0) + log(glucoseT0))[78].

**Liver histology**. For hematoxylin and eosin (H&E) staining, the livers were frozen in OCT embedding matrix and on dry ice. Sectioning and staining were done according to standard histological procedures.

**Biochemical analysis of plasma**. Within 15 min after blood collection, plasma was separated by centrifugation (15 min at 2000 RPM). Plasma total triglycerides (Total TG) and total cholesterol (Total Chol) were measured by enzymatic assay using commercially available kits (Roche Diagnostics GmbH, Mannheim and mti Diagnostic GmbH, Idstein, Germany). Cholesterol lipoprotein fractions in serum were determined as described[79]. Briefly, sera from each individual mouse were separated by size exclusion chromatography using a Superose and PC 3.2/30 column (Pharmacia Biotech, Uppsala, Sweden). Reagent (Roche Diagnostic, Mannheim, Germany) was directly infused into the eluate online and the absorbance was measured. The concentration of the different lipoprotein fractions was calculated from the area under the curves of the elution profiles by using the EZChrom Elite software (Scientific Software; Agilent Technologies, Santa Clara, CA).

**Immunoassay for adipokine levels**. Within 15 min after blood collection, plasma was separated by centrifugation (15 min at 2000 RPM). A Multiplexed bead immunoassay was used to measure adipokine levels using a commercially available kit (Bio-Plex Pro Mouse Diabetes 8-Plex Assay #171F7001M) according to manufacturer's instructions.

### Lipidomic

*Fatty acid analysis using gas chromatography–mass spectrometry (GC-MS)*. Total lipid extracts were obtained using a modified Bligh and Dyer method[80] and after transmethylation, the fatty acids were analyzed by gas chromatography followed by mass spectrometry (GC-MS)[81,82]. Aliquots of the lipid extracts corresponding to 2.5 µg of total phospholipid, were transferred into glass tubes and dried under a nitrogen stream. Resulting lipid films were dissolved in 1 mL of *n*-hexane containing a C19:0 as internal standard (1.03 µg mL$^{-1}$, CAS number 1731-94-8, Merck, Darmstadt, Germany) with addition of 200 µL of a solution of potassium hydroxide (KOH, 2 M) in methanol, followed by 2 min vortex. Then, 2 mL of a saturated solution of sodium chloride (NaCl) was added, and the resulting mixture was centrifuged for 5 min at $626 \times g$ for phase separation. Cholesterol was removed from the organic phase according to the Lipid Web protocol (https://lipidhome.co.uk/ms/basics/msmeprep/index.htm). A 1 cm silica column in a pipette tip with wool was pre-conditioned with 5 mL of hexane (high-performance liquid chromatography (HPLC) grade). Methyl esters were added to the top of the tip and recovered by elution with hexane:diethyl ether (95:5, v/v, 3 mL), and thereafter dried under a nitrogen current. Fatty acid methyl esters (FAMEs) were dissolved in 100 µL, and 2.0 µL were injected in GC-MS (Agilent Technologies 8860 GC System, Santa Clara, CA, USA). GC-MS was equipped with a DB-FFAP column (30 m long, 0.32 mm internal diameter, and 0.25 µm film thickness (J & W Scientific, Folsom, CA, USA)). The GC equipment was connected to an Agilent 5977B Mass Selective Detector operating with an electron impact mode at 70 eV and scanning the range $m/z$ 50–550 in a 1 s cycle in a full scan mode acquisition. Oven temperature was programmed from an initial temperature of 58 °C for 2 min, a linear increase to 160 °C at 25 °C min$^{-1}$, followed by linear increase at 2 °C min$^{-1}$ to 210 °C, then at 20 °C min$^{-1}$ to 225 °C, standing at 225 °C for 20 min. Injector and detector temperatures were set to 220 and 230 °C, respectively. Helium was used as the carrier gas at a flow rate of 1.4 mL min$^{-1}$. GCMS5977B/Enhanced Mass Hunter software was used for data acquisition. To identify fatty acids, the acquired data were analyzed using the qualitative data analysis software Agilent MassHunter Qualitative Analysis 10.0. Fatty acids identification was performed by MS spectrum

comparison with the chemical database NIST library and confirmed with the literature.

*Phospholipids (PL), sphingolipids (SL), and triglycerides (TG) analysis by liquid chromatography–mass spectrometry.* Total lipid extracts from the left lobe of the liver were separated using a HPLC system (Ultimate 3000 Dionex, Thermo Fisher Scientific, Bremen, Germany) with an autosampler coupled online to a Q-Exactive hybrid quadrupole Orbitrap mass spectrometer (Thermo Fisher Scientific, Bremen, Germany), adapted from[80,83]. Briefly, the solvent system consisted of two mobile phases: mobile phase A (ACN/MeOH/water 50:25:25 (v/v/v) with 2.5 mM ammonium acetate) and mobile phase B (ACN/MeOH 60/40 (v/v) with 2.5 mM ammonium acetate). Initially, 10% of mobile phase A was held isocratically for 2 min, followed by a linear increase to 90% of A within 13 min and a maintenance period of 2 min, returning to the initial conditions in 3 min, followed by a re-equilibration period of 10 min prior to the next injection. Five μg of phospholipid (PL) from total lipid extracts were mixed with 4 μL of phospholipid standard mixture (dMPC—0.02 μg, dMPE—0.02 μg, SM—0.02 μg, LPC—0.02 μg, TMCL—0.08 μg, dPPI—0.08 μg, dMPG—0.012 μg, dMPS—0.04 μg, Cer—0.04 μg, dMPA—0.08 μg) and 91 μL of solvent system (90% of eluent B and 10% of eluent A). Five μL of each dilution were introduced into the AscentisSi column (10 cm × 1 mm, 3 μm, Sigma-Aldrich, Darmstadt, Germany) with a flow rate of 50 μL min$^{-1}$. The temperature of the column oven was maintained at 35 °C. The mass spectrometer with Orbitrap technology operated in positive (electrospray voltage 3.0 kV) and negative (electrospray voltage −2.7 kV) ion modes with a capillary temperature of 250 °C, a sheath gas flow of 15 U, a high resolution of 70,000 and AGC target 1e6. In MS/MS experiments, cycles consisted of one full scan mass spectrum and ten data-dependent MS/MS scans (resolution of 17,500 and AGC target of 1e5), acquired in each polarity. Cycles were repeated continuously throughout the experiments with the dynamic exclusion of 60 s and an intensity threshold of 2e4. Normalized collisional energy ranged between 20, 25, and 30 eV.

*Reagents/chemicals for LC-MS analysis.* Phospholipid internal standards 1,2-dimyristoyl-*sn*-glycero-3-phosphocholine (dMPC), 1-nonadecanoyl-2-hydroxy-*sn*-glycero-3-phosphocholine (LPC), 1,2-dimyristoyl-*sn*-glycero-3-phosphoethanolamine (dMPE), N-palmitoyl-D-*erythro*-sphingosylphosphorylcholine (NPSM – SM d18:1/17:0), N-heptadecanoyl-D-erythro-sphingosine (Cer d18:1/17:0), 1,2-dimyristoyl-*sn*-glycero-3-phospho-(10-rac-)glycerol (dMPG), 1,2-dimyristoyl-*sn*-glycero-3-phospho-L-serine (dMPS), tetramyristoylcardiolipin (TMCL), 1,2-dimyristoyl-*sn*-glycero-3-phosphate (dMPA), and 1,2-dipalmitoyl-*sn*-glycero-3-phosphatidylinositol (dPPI) were purchased from Avanti Polar Lipids, Inc. (Alabaster, AL). HPLC grade dichloromethane, methanol, and acetonitrile were purchased from Fisher Scientific (Leicestershire, UK). All the reagents and chemicals used were of the highest grade of purity commercially available and were used without further purification. The water was of Milli-Q purity (Synergy®, Millipore Corporation, Billerica, MA).

Spectra were analyzed in positive and negative mode, depending on the lipid class. Ceramides (Cer), glucosylceramides (GlcCer), phosphatidylethanolamine (PE), lyso phosphatidylethanolamine (LPE), phosphatidylcholine (PC), lysophosphatidylcholine (LPC), and sphingomyelin (SM) were analyzed in the LC-MS spectra in the positive ion mode, and identified as [M + H]$^+$ ions, while cardiolipin (CL), phosphatidylserine (PS), phosphatidylinositol (PI), lysophosphatidylinositol (LPI) and phosphatidylglycerol (PG) species were analyzed in negative ion mode, and identified as [M − H]$^-$ ions. Molecular species of triacylglycerol (TG) were also analyzed in positive ion mode as [M + NH$_4$]$^+$ ions. Data acquisition was carried out using the Xcalibur data system (V3.3, Thermo Fisher Scientific, USA). The mass spectra were processed and integrated through the MZmine software (v2.32)[84]. This software allows for filtering and smoothing, peak detection, alignment and integration, and assignment against an in-house database, which contains information on the exact mass and retention time for each PL, Cer, and TG molecular species. During the processing of the data by MZmine, only the peaks with raw intensity higher than 1e4 and within 5 ppm deviation from the lipid exact mass were considered. The identification of each lipid species was validated by analysis of the LC-MS/MS spectra. The production at *m/z* 184.07 (C$_5$H$_{15}$NO$_4$P), corresponding to phosphocholine polar head group, observed in the MS/MS spectra of the [M + H]$^+$ ions allowed to pinpoint the structural features of PC, LPC, and SM molecular species under MS/MS conditions[80], which were further differentiated based on *m/z* values of precursor ions and characteristic retention times. PE and LPE molecular species ([M + H]$^+$ ions) were identified by MS/MS based on the typical neutral loss of 141 Da (C$_2$H$_8$NO$_4$P), corresponding to phosphoethanolamine polar head group. These two classes were also differentiated based on *m/z* values of precursor ions and characteristic retention times. The [M + H]$^+$ ions of Cer and GlcCer molecular species were identified by the presence of product ions of sphingosine backbone in MS/MS spectra, such as ions at *m/z* 264.27 (C$_{18}$H$_{34}$N) and 282.28 (C$_{18}$H$_{36}$NO) for sphingosine d18:1[85], together with the information on *m/z* values of precursor ions and characteristic retention times. The PG molecular species were identified by the [M − H]$^-$ ions and based on the product ions identified in the corresponding MS/MS spectra, namely the product ions at *m/z* 152.99 (C$_3$H$_6$O$_5$P) and 171.01 (C$_3$H$_8$O$_6$P). PI and LPI, also identified as [M−H]$^-$ ions, were confirmed the product ions at *m/z* 223.00 (C$_6$H$_8$O$_7$P), 241.01 (C$_6$H$_{10}$O$_8$P), 297.04 (C$_9$H$_{14}$O$_9$P), and 315.05 (C$_9$H$_{16}$O$_{10}$P), which all derived from phosphoinositol polar head group[80,86]. The [M − H]$^-$ ions of PS molecular species

were identified based on product ions at *m/z* 152.99 (C$_3$H$_6$O$_5$P) in MS/MS spectra, retention time and *m/z* values of precursor ions. CL molecular species ([M − H]$^-$ ions) were characterized by MS/MS with identification of ions at *m/z* 152.99 (C$_3$H$_6$O$_5$P), carboxylate anions of fatty acyl chains (RCOO$^-$), product ions corresponding to phosphatidic acid anion and phosphatidic acid anion plus 136 Da[86]. Negative ion mode MS/MS data were used to identify the fatty acid carboxylate anions RCOO$^-$, which allowed the assignment of the fatty acyl chains esterified to the PL precursor. The MS/MS spectra of [M + NH$_4$]$^+$ ions of TGs allowed the assignment of the fatty acyl substituents on the glycerol backbone[87].

**Unsupervised clustering.** The raw data matrix of the lipid spectra was distributed column-wise by sample IDs and row-wise by PL names. The TMM method was used to normalize between samples[88]. Unsupervised clustering was then performed using the principal component analysis (PCA) plot option in R. The PCA plot is based on the two most variant dimensions in which the PL parameters with duplicated data are filtered out.

Similarly, for the Uniform Manifold Approximation and Projection (UMAP) plot, the raw data matrix was distributed column-wisely by sample ID and row-wisely by gene names. Unsupervised clustering was then performed using the UMAP plot function in R. The plot is based on the two most variant dimensions of UMAP, in which the genes with duplicated reads are filtered out.

**RNA isolation, purity, and integrity determination.** Liver, SAT and VAT total RNA was extracted using QIAGEN miRNeasy Mini Kit (217004, Qiazol). RNA concentration was measured by nanodrop®. RNA was treated with RNase-free DNase (79254) according to the manufacturer's instructions. cDNA libraries were prepared for bulk-RNA sequencing as detailed[15].

**Bulk RNA-seq mapping.** All raw sequence reads available in FastQ format were mapped to the mouse genome (mm10) using Tophat2 with Bowtie2 option[89,90], as described[15]. Raw read counts for each gene were calculated using featureCounts from the subread package[91].

**Bulk RNA-seq differential gene expression analysis.** A differential gene expression analysis was performed using DEseq2[92]. The differentially expressed (DE) genes were identified by adjusted *p*-value for multiple testing using Benjamini–Hochberg correction with false discovery rate (FDR) values <0.1.

**Pathway analysis.** A Gene Set Enrichment Analysis (GSEA) was performed using the KEGG pathways dataset. Genes were ranked in descending order according to the log$_2$ fold change (log$_2$FC) of expression. Differences between the ranks of genes in a pathway were compared to other genes. For each queried pathway, if gene *i* is a member of the pathway, it is defined as:

$$Xi = \sqrt[2]{\frac{N-G}{G}} \qquad (1)$$

if gene *i* is not a member of the pathway, it is defined as:

$$Xi = -\sqrt[2]{\frac{G}{N-G}} \qquad (2)$$

here *N* is the total number of genes and *G* indicates the number of genes in the query pathway. Next, a max running sum across all *N* genes Maximum Estimate Score (MES) is calculated as:

$$MES = \max_{1 \le j \le N} \sum_{i=1}^{j} Xi \qquad (3)$$

The permutation test was performed 1000 times to judge the significance of MES values. The queried pathway with a nominal *p*-value less than 0.05 and FDR values <0.1 were considered significantly enriched. The positive MES value indicates enrichment (upregulation) whereas a negative MES value indicates depletion (downregulation) of a pathway activity.

**Gene Ontology (GO) enrichment analysis.** Gene Ontology enrichment analysis was performed using online software AmiGO website (http://amigo.geneontology.org/amigo), where the significant enrichment GO terms was identified using Fisher's Exact test with *P* values ≤0.05.

**Statistics and reproducibility.** The details about experimental design and statistics used in different data analyses performed in this study are given in the respective sections of results and methods. Data are expressed as mean ± sem. Differences between the four group (female and male offspring (sex) and C and HF (diet) mother groups (F-moC, M-moC, F-moHF, and M-momHF) were determined using two-way ANOVA with diet (D) and sex (S) as independent variables, followed by Tukey's multiple comparison post hoc test when significant (*p* < 0.05). Differences between two groups (sexes, F versus M; maternal diet moC versus moHF) were determined by unpaired two-tailed *t*-test corrected for multiple comparisons using the Holm–Sidak method, with alpha = 5.000%. Statistical

analysis and graphs were generated using Graphpad prism v7.1.2 software. For RNAseq data analysis, please refer to the method section for each analysis. *, $p < 0.05$ M versus F and #, $p < 0.05$ moHF versus moC were considered significant. ** or ##, $p < 0.01$; *** or ###, $p < 0.001$.

**Reporting summary**. Further information on research design is available in the Nature Research Reporting Summary linked to this article.

## Data availability

The raw data generated for lipidomics, and RNA sequencing are available as described below: For VAT and SAT RNA sequencing, SRA data: PRJNA662930. For Liver RNA sequencing, SRA data: PRJNA723771. For the lipidomic, SRA data: https://figshare.com/s/ac91b57eaa0f5c560d3d. Raw data for magnetic resonance imaging and spectroscopy were generated at the Experimental Research and Imaging Centre at Karolinska University Hospital, Solna, Sweden facility. Derived data supporting the findings of this study are available from the corresponding author upon request. All other data are available from the corresponding author on reasonable request.

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

## Acknowledgements

The MRI and MRS experiments were performed at the Department of Comparative Medicine/Karolinska Experimental Research and Imaging Centre at Karolinska University Hospital, Solna, Sweden. We thank Peter Damberg and Sahar Nikkhou Aski for excellent assistance to develop the sequence for proton-magnetic resonance spectroscopy in the liver. We thank Ingela Arvidsson for excellent help at the FLPC for lipoprotein profiling. We thank Byambajav Buyandelger, Sonja Gustafsson, Jianping Liu from the single cell facility, Karolinska institute in Huddinge for excellent assistance for the SmartSeq2 experiment. This work and M.K.A. were supported by the Novo Nordisk Foundation (NNF14OC0010705), the Lisa and Johan Grönbergs Foundation (2019-00173) and by AstraZeneca (ICMC). C.K. is supported by the Knut & Alice Wallenberg foundation (KAW 2016.0174) and the Swedish Research Council (2019–05165). L.A.H. is supported by grants from FCT - Fundação para a Ciência e a Tecnologia (UID/BIM/04501/2020), CCDRC (CENTRO-01-0145-FEDER-000003) and CCDRC (CENTRO-01-0246-FEDER-000018). M.R.D., D.C., and T.M. are supported by CESAM (UIDP/50017/2020+UIDB/50017/2020) and LAQV/REQUIMTE (UIDB/50006/2020). B.O. was supported by grants from the Swedish Heart-Lung Foundation, the Region Stockholm/Karolinska Institutet (ALF), CIMED, and AstraZeneca (ICMC). Fetus in Fig. 1a was created by Servier Medical Art "newborn mouse". In Fig. 6, lipoprotein cells were designed using Servier Medical Art "lipids" and the artery was extracted from Servier Medical Art "capillary compartment" image, http://smart.servier.com/. Open Access licensed under a Creative Common Attribution 3.0 Generic License https://creativecommons.org/licenses/by/3.0/legalcode.

## Author contributions

M.K.A. conceptualized and designed the study. C.S., M.G.G., and M.K.A. performed animal experiments; C.S. and M.K.A. collected and analyzed all generated data; L.H., D.C., T.M., and M.R.D. performed the lipidomics and wrote the method for lipidomic; C.S. performed RNA sequencing experiments; C.S. and X.L. performed the bioinformatics; C.S. and M.K.A. designed the figures and wrote the manuscript; L.H., B.A., and C.K. substantially participated to the manuscript review. The manuscript was edited and approved by all authors. M.K.A. is the guarantor of this work and, as such, had full access to all the data in the study

and takes responsibility for the integrity of the data and the accuracy of the data analysis. All authors approved the final version of the manuscript.

## Funding

## Competing interests
The authors declare no competing interests.
