## [Peer Review File · Communications Biology]

Reviewers' comments:

Reviewer #1 (Remarks to the Author):

Maternal obesity is a relevant health issue in many countries nowadays. It has been shown in many epidemiological but also animal studies that the offspring is programmed to metabolic disease on the long run. This has also been shown to be sex-dependent, with males usually being the more susceptible sex.

Thus, the authors here study the consequences of maternal high-fat feeding and obesity on the offspring's liver. The experiments are clearly structured. However, neither the experimental setup nor the analyses done are novel but have been performed in similar ways by several laboratories in the last decade, although not to this level of detail. Also the findings are interesting but largely confirmative.

I see two major problems in this study. First, the dams are fed control or high fat diet until weaning, leading to two groups of offspring. However, in the next step all offspring is weaned to high fat diet. Therefore, a comparison to a "real" control group (control diet in pregnancy and control diet for the offspring) is not possible. This limits the value of the experiments. Second, the authors focus on the long-term outcome in relation to sex, which should be attributed to hormonal differences. This neglects the possibility that already the initial programming step (in utero) might be sex dependent. This is especially disappointing as they mention "in utero" in the title of the manuscript. It would have been extremely interesting to see data on these initial steps induced by the high fat diet.

Reviewer #2 (Remarks to the Author):

Dear authors,

Your study presents an extensive analysis of the effects of maternal obesity in a rodent model and adds a great value to what is already known with regards to sex-specific programming, and sex differences in normal physiology and disease. Combination of various methods used makes your study comprehensive. Especially notable is a thorough lipidomic analysis. For the reviewer results presented are reliable and accountable, however, before publication some focal points should be addressed:

Major comments

1. Lines 75-76 'However, the mechanisms by which MO might differently program transcriptional and biological activities in female and male offspring have not been assessed' – this statement is not accurate studies on MO effects on female and male offspring have been carried before. For example: Chang, E., Hafner, H., Varghese, M. et al. Programming effects of maternal and gestational obesity on offspring metabolism and metabolic inflammation. *Sci Rep* 9, 16027 (2019). <https://doi.org/10.1038/s41598-019-52583-x>
2. In results section authors should consider stating P or adjusted q values (FDR)
3. For supplementary tables authors should state FDR and nominal P values as well as fold changes of genes
4. Authors state that "Altogether these results indicated that MO alters the liver transcriptome to a much larger extent in females than in males" – however, authors should consider if number of animals analysed (6 females vs 3 males) could have an effect on the outcome of the analysis?
5. For RNA isolation, purity and integrity determination – authors should state how was the integrity determined and state RIN numbers
6. Can authors justify why an FDR of 0.1 rather than 0.05 was used for pathway analysis? How many of genes analysed has passed a cut-off of FDR 0.05?

Minor comments

1. Line 85 – comma missing after TG accumulation
2. Line 97 – BW abbreviation needs to be explained here
3. Line 105 – abbreviation maternal obesity (MO) has been introduced earlier
4. Line 108-110 'At MID, F-moHF had higher ratio of total fat (TF) on BW than M-moHF and M-moHF but not F-moHF accumulated less fat compared to moC at MID which vanished at END – sentence really difficult to understand.
5. Line 117-122 – 'Males showed reduced glucose tolerance and insulin sensitivity (high fasting insulin level and during OGTT) compared to females; but females only showed reduced insulin sensitivity by MO during the OGTT (higher insulin levels) (Figs.1h-1i). Surprisingly, MO do not alter glucose disappearance after insulin injection (Fig.1j), which indicated pancreatic disturbances in F-moHF during the OGTT – sentences difficult to understand – consider rewording.
6. Line 126 – in all conditions – specify which conditions
7. Line 127 – physiological changes – can you specify?
8. Line 146 – 'highe' typo
9. Line 140 – 141 'The four DE genes Pdk1, Lpin1, Prlr and Nox4 have been known to act as key regulators of hepatic insulin sensitivity' – can you provide reference?
9. Line 148-149 – 'Interestingly, hepatic ERS1 has been shown to regulate G6pc and Pck1 and to be critical in the regulation of glucose metabolism – can you provide reference?
10. Figure 1q (PCK1) – not consistent with other figures
11. Lines 305-307 – 'Interestingly, MO caused the downregulation of the Nsdhl, G6pdx, Rbm3, Ctps2 and Acls4 genes, which are involved in lipid and cholesterol metabolism and the upregulation of the Apex2 gene that is linked to hepatocarcinoma development' – can you provide references?
12. Line 308 – Apex2 and Pgrmc1 – are these genes only involved in hepatocarcinoma? Is there another role/explanation?
13. Lines 313-315 – 'We observed that Il13ra1 and Kdm5c encoding for liver lipid homeostasis were higher expressed in females, and Atp11c and Acls4 encoding for genes connected to lipid disorders and hepatocarcinoma development were higher expressed in males – can you provide references?
14. Lines 355-356 – 'reduced expression levels of RNA Binding Motif Protein (Rbm)3, CTP synthase (Ctps)2 and Acls4 X-linked genes involved in hepatocellular carcinoma development' – can you provide references?
15. Line 368-369 – 'Remarkably, we found two key genes (Osgin1 and Stat1) that are known tumor repressors – can you provide reference?
16. Line 402 – 'at early life stage of life' – repetition
17. Lines 516-517 – 'for HFD mother and HFD offspring) and the group of offspring born from CD fed mother named moC (for CD mother and HFD offspring) – is that correct?

Reviewers' comments:

We appreciate the time and effort that the editor and the reviewers dedicated to providing feedback on our manuscript. We would like to thank the reviewers for their thoughtful comments and efforts towards improving our manuscript. Please find our point-by-point response to the reviewers' concerns below.

Reviewer #1 (Remarks to the Author):

Maternal obesity is a relevant health issue in many countries nowadays. It has been shown in many epidemiological but also animal studies that the offspring is programmed to metabolic disease on the long run. This has also been shown to be sex-dependent, with males usually being the more susceptible sex.

Thus, the authors here study the consequences of maternal high-fat feeding and obesity on the offspring's liver. The experiments are clearly structured. However, neither the experimental setup nor the analyses done are novel but have been performed in similar ways by several laboratories in the last decade, although not to this level of detail. Also the findings are interesting but largely confirmative.

I see two major problems in this study. First, the dams are fed control or high fat diet until weaning, leading to two groups of offspring. However, in the next step all offspring is weaned to high fat diet. Therefore, a comparison to a "real" control group (control diet in pregnancy and control diet for the offspring) is not possible. This limits the value of the experiments. Second, the authors focus on the long-term outcome in relation to sex, which should be attributed to hormonal differences. This neglects the possibility that already the initial programming step (in utero) might be sex dependent. This is especially disappointing as they mention "in utero" in the title of the manuscript. It would have been extremely interesting to see data on these initial steps induced by the high fat diet.

First: "real" control group

Answer: As mentioned in the introduction (line 64) and later discussed (line 413), we have indeed a control group with mothers and postweaning offspring on CD. However, we have already published in a separate paper the results (PMID: 33398027); the current manuscript being a follow-up study. In the current study we aimed to investigate the effect of maternal HFD on offspring metabolic profile when offspring are exposed to an obesogenic post-weaning diet (as often the case in humans). Further, we aimed to investigate whether the early life exposure (in utero and lactation) to HFD (compared to the CD) could accelerate the adverse effects of the post-weaning obesogenic diet in offspring. Therefore, we assumed using the group of offspring born from control diet mothers (moC) as our "control group" in the current manuscript.

Second: long term outcomes and sex hormones

Answer: we agree that the long-term outcome in relation to sex could partly be explained by the differences in sex hormones and not solely to in utero programming by the maternal diet. However, a recent paper shows sex differences in the metabolic response to maternal obesity at late gestation (E18.5) (PMID: 35025731). Significant differences in the heart lipidome were observed in female fetuses compared to male fetuses born from obese mothers. Another study in humans has shown that exposure to maternal obesity causes sex-dependent alterations in miRNA and gene expression in human fetal liver (PMID: 31852997).

It is rather fascinating how these differences between the sexes can occur given that fetuses of both sexes are exposed to the same maternal metabolic milieu. It is not clear if these sex-specific responses early in life could define an ability of females to adapt to the environment and to protect against longer term detrimental effects. This evidence shows that the intrauterine environment (exposure to obesogenic environment) during the period of cellular differentiation and growth might result in changes in cellular functions that will affect female and male offspring differently in the long term, regardless of their sexual hormones. However, the exact mechanisms that explain the sexual dimorphic response to maternal obesity remains still elusive.

Reviewer #2 (Remarks to the Author):

Dear authors,

Your study presents an extensive analysis of the effects of maternal obesity in a rodent model and adds a great value to what is already known with regards to sex-specific programming, and sex differences in normal physiology and disease. Combination of various methods used makes your study comprehensive. Especially notable is a thorough lipidomic analysis. For the reviewer results presented are reliable and accountable, however, before publication some focal points should be addressed:

Major comments

1. Lines 75-76 'However, the mechanisms by which MO might differently program transcriptional and biological activities in female and male offspring have not been assessed' – this statement is not accurate studies on MO effects on female and male offspring have been carried before. For example: Chang, E., Hafner, H., Varghese, M. et al. Programming effects of maternal and gestational obesity on offspring metabolism and metabolic inflammation. Sci Rep 9, 16027 (2019). <https://doi.org/10.1038/s41598-019-52583-x>

Answer: We have modified the sentence to stress the novelty of the combination of various techniques used in the current manuscript (line 73-75) "While sex dependent metabolic adaptation in response to MO have been described (<https://doi.org/10.1038/s41598-019-52583-x>) the mechanisms by which MO might differently program hepatic lipidome and transcriptional activity in female and male offspring have not been assessed".

2. In results section authors should consider stating P or adjusted q values (FDR)

Answer: We have added this information in the result section. Line 125 "We performed RNA-seq and considered significantly expressed genes and pathways with FDR <0.1 and p-value <0.05."

3. For supplementary tables authors should state FDR and nominal P values as well as fold changes of genes

Answer: All presented genes have a p-value <0.05 and FDR <0.1 as stated on the table legend. The Fastq files have been uploaded in SRA data: PRJNA723771. However, we have added as supplementary table the DE genes analysis with the FDR and Log2Fold change (TableS1_DE genes analysis).

4. Authors state that "Altogether these results indicated that MO alters the liver transcriptome to a much larger extent in females than in males' – however, authors should consider if number of animals analysed (6 females vs 3 males) could have an effect on the outcome of the analysis?"

Answer: We agree with the reviewer that the differences in the number of animals used for the analysis may have an effect on the outcome. Therefore, we made some additional analysis to look at the homogeneity of our group (tSNE plot).

1. In the unsupervised tSNE clustering plot, the six female samples clearly separate from the three male samples
2. When we use 6 female vs 3 male, the degree of freedom of wald test in Deseq2 is $(6-1) + (3-1) = 7$. When we randomly select 3 female vs 3 male, the degree of freedom of wald test in Deseq2 is $(3-1) + (3-1) = 4$. Under the same expression values, the tested genes with higher degree of freedom tend to be more significant than the ones with lower degree of freedom.
3. Therefore we chose to select six females to increase the degree of freedom to avoid the possible false negative results and also account for outliers.

5. For RNA isolation, purity and integrity determination – authors should state how was the integrity determined and state RIN numbers

Answer: For the Smart-seq2 experiment (RNA-seq) we focused mostly on the quality of cDNA rather than the quality of RNA since its more important to get a good quality cDNA in order to continue with the sequencing. However, we checked the RIN values of the RNA, and we found an average number of 7.6.

6. Can authors justify why an FDR of 0.1 rather than 0.05 was used for pathway analysis? How many of genes analysed has passed a cut-off of FDR 0.05?

Answer: We used a cut-off of $FDR < 0.1$ to reduce the false negative in our analysis, we did not want to miss any potential causative genes and pathways. Nevertheless, we extracted the DE genes with a cut-off of FDR of 0.05 to make a comparison. Below is the Venn diagram we generated with the genes that passed the cut-off of FDR 0.1 (yellow circle) and FDR 0.05 (blue circle) for the four comparisons. As expected, all genes with a cut-off of FDR at 0.05 are included into the set of genes that are found with a cut-off of FDR at 0.1, without major differences, except for the M-moC versus M-moHF comparison were almost 50% of the DE genes were lost with an FDR cut-off of 0.05 compared to 0.1 (but a very small number of genes are deregulated in males anyway).

Minor comments

1. Line 85 – comma missing after TG accumulation

Done

2. Line 97 – BW abbreviation needs to be explained here

Done

3. Line 105 – abbreviation maternal obesity (MO) has been introduced earlier

Done

4. Line 108-110 'At MID, F-moHF had higher ratio of total fat (TF) on BW than M-moHF and M-moHF but not F-moHF accumulated less fat compared to moC at MID which vanished at END – sentence really difficult to understand.

Answer: We have modified the sentence and we hope this is now clearer for the reader. Line 106: "At MID, M-moHF had lower ratio of total fat (TF) on BW than F-moHF due to reduction of fat mass compared to M-moC; this difference disappeared at END"

5. Line 117-122 – 'Males showed reduced glucose tolerance and insulin sensitivity (high fasting insulin level and during OGTT) compared to females; but females only showed reduced insulin sensitivity by MO during the OGTT (higher insulin levels) (Figs.1h-1i). Surprisingly, MO do not alter glucose disappearance after insulin injection (Fig.1j), which indicated pancreatic disturbances in F-moHF during the OGTT – sentences difficult to understand – consider rewording.

Answer: we have rewritten the sentence to clarify the result. Line 115-119: "Males showed reduced glucose tolerance and insulin sensitivity (high fasting insulin level and during OGTT) compared to females. However, only females showed reduced insulin sensitivity by MO during the OGTT (higher insulin levels) (Figs.1h-1i and 1k-1l). Interestingly, insulin tolerance test showed that MO did not alter glucose disappearance in F-moHF (Fig.1j)."

6. Line 126 – in all conditions – specify which conditions

Answer: We have changed the sentence. Line 121: "The quantitative insulin-sensitivity check index (QUICKI) indicated that males were more insulin resistant than females at both MID and END, regardless of maternal diet.

7. Line 127 – physiological changes – can you specify?

Answer: we have added information to clarify the physiological changes. Line 123: "We tested whether the physiological changes described above (*in vivo* data) were accompanied by changes in gene expression in the liver."

8. Line 146 – 'highe' typo

Done

9. Line 140 – 141 'The four DE genes Pdk1, Lpin1, Prlr and Nox4 have been known to act as key regulators of hepatic insulin sensitivity' – can you provide reference?

Answer: We have added references for the genes listed in the results section. Line 138: "The four DE genes *Pdk1*, *Lpin1*, *Prlr* and *Nox4* are key regulators of hepatic insulin sensitivity¹⁻⁴, and were altered by sex and/or by MO." Pdk1 (PMID: 15554902), Prlr (PMID: 23775766), Nox4 (PMID: 22328777), Lpin1 (PMID: 19254569)

9. Line 148-149 – 'Interestingly, hepatic ERS1 has been shown to regulate G6pc and Pck1 and to be critical in the regulation of glucose metabolism – can you provide reference?

Answer: We have added references for the genes listed in the results section. Line 145: “Interestingly, hepatic ESR1 has been shown to regulate *G6pc*⁵ and *Pck1*⁶ and to be critical in the regulation of glucose metabolism” ESR1 + *G6pc* (PMID: 32150359), ESR1 + *Pck1* (PMID: 28490809)

10. Figure 1q (PCK1) – not consistent with other figures

Answer: We have modified the Fig.1q to have it consistent with the other figures.

11. Lines 305-307 – ‘Interestingly, MO caused the downregulation of the *Nsdhl*, *G6pdx*, *Rbm3*, *Ctps2* and *Acls4* genes, which are involved in lipid and cholesterol metabolism and the upregulation of the *Apex2* gene that is linked to hepatocarcinoma development’ – can you provide references?

Answer: We have provided the references for each of these genes. Line 300-305: “In females, MO reduced the expression of *Nsdhl* and *Acls4* genes, which are involved in lipid and cholesterol metabolism^{7,8}. Interestingly, MO altered expression of *G6pdx*, *Rbm3*, *Ctps2* and *Apex2* genes, that are linked to hepatocarcinoma development⁹⁻¹². In males, MO reduced expression of *Zdhhc9* gene, involved in cancer and metabolism¹³ (Figs.4f-4g). Importantly, expression of X-linked genes *Apex2*, *Acls4* and *Pgrmc1*, involved in lipid metabolism and hepatocarcinoma was sex-dependent^{8,9,14} .”

Nsdhl (PMID: 14506130), *G6pdx* (PMID: 33375092), *Apex2* (PMID: 32775374), *Rbm3* (PMID: 31235426), *Ctps2* (PMID: 29097181), *Acls4* (PMID: 33340617).

12. Line 308 – *Apex2* and *Pgrmc1* – are these genes only involved in hepatocarcinoma? Is there another role/explanation?

Answer: We have provided additional references about the role of *Apex2* and *Pgrmc1* genes. Line 304: “Importantly, expression of X-linked genes *Apex2*, *Acls4* and *Pgrmc1*, involved in lipid metabolism and hepatocarcinoma was sex-dependent^{8,9,14}.” *PGRMC1* is involved in many metabolic pathways including fatty liver amelioration (PMID: 30356113), cytochrome activities (PMID: 21825115), drug metabolism, cholesterol synthesis, and steroid synthesis (PMID: 18992768) and hepatic glucose metabolism (PMID: 34970218).

13. Lines 313-315 – ‘We observed that *Il13ra1* and *Kdm5c* encoding for liver lipid homeostasis were higher expressed in females, and *Atp11c* and *Acls4* encoding for genes connected to lipid disorders and hepatocarcinoma development were higher expressed in males – can you provide references?’

Answer: We have provided the references for each of these genes Line: 307-310 “Expression of *Il13ra1* and *Kdm5c* encoding for liver glucose¹⁵ and lipid¹⁶ homeostasis was higher in females, and expression of *Atp11c* and *Acls4* encoding for genes connected to lipid disorders and hepatocarcinoma development^{8,17} was higher in males.” *Il13ra1* (PMID: 23257358), *Acls4* (PMID: 33340617), *Atp11c* (PMID: 30018401) and *Kdm5c* (PMID: 32714863)

14. Lines 355-356 – ‘reduced expression levels of RNA Binding Motif Protein (*Rbm3*), CTP synthase (*Ctps2*) and *Acls4* X-linked genes involved in hepatocellular carcinoma development’ – can you provide references?’

Answer: We have provided the references for each of these genes. Line 348: “MO reduced these areas of cell proliferation in females, which is in line with reduced expression levels of *Rbm3*, *Ctps2* and *Acls4* X-linked genes^{8,10,11}” *Rbm3* (PMID: 31235426), *Ctps2* (PMID: 29097181), *Acls4* (PMID: 33340617)

15. Line 368-369 – ‘Remarkably, we found two key genes (*Osgin1* and *Stat1*) that are known tumor repressors – can you provide reference?’

Answer: We have provided the references for each of these genes. Line 361: "Remarkably, we found two key genes (*Osgin1* and *Stat1*) that are known tumor repressors^{18,19}." *Osgin1* (PMID: 24417816), *Stat1*(PMID: 23588992)

16. Line 402 – 'at early life stage of life' – repetition
Done

17. Lines 516-517 – 'for HFD mother and HFD offspring) and the group of offspring born from CD fed mother named moC (for CD mother and HFD offspring) – is that correct?

Answer: We have modified the sentence to clarify the group names. Line 500-502: "To simplify the naming convention, the group of offspring born from HFD fed dams were named "moHF", and the group of offspring born from CD fed dams were named "moC""

Added references

- 1 Mora, A., Lipina, C., Tronche, F., Sutherland, C. & Alessi, D. R. Deficiency of PDK1 in liver results in glucose intolerance, impairment of insulin-regulated gene expression and liver failure. *Biochem J* **385**, 639-648, doi:10.1042/BJ20041782 (2005).
- 2 Yu, J. *et al.* PRLR regulates hepatic insulin sensitivity in mice via STAT5. *Diabetes* **62**, 3103-3113, doi:10.2337/db13-0182 (2013).
- 3 Wu, X. & Williams, K. J. NOX4 pathway as a source of selective insulin resistance and responsiveness. *Arterioscler Thromb Vasc Biol* **32**, 1236-1245, doi:10.1161/ATVBAHA.111.244525 (2012).
- 4 Ryu, D. *et al.* TORC2 regulates hepatic insulin signaling via a mammalian phosphatidic acid phosphatase, LIPIN1. *Cell Metab* **9**, 240-251, doi:10.1016/j.cmet.2009.01.007 (2009).
- 5 Lundholm, L. *et al.* The estrogen receptor alpha-selective agonist propyl pyrazole triol improves glucose tolerance in ob/ob mice: potential molecular mechanisms. *J Endocrinol* **243**, X1, doi:10.1530/JOE-08-0192e 10.1677/JOE-08-0192 (2019).
- 6 Qiu, S. *et al.* Hepatic estrogen receptor alpha is critical for regulation of gluconeogenesis and lipid metabolism in males. *Sci Rep* **7**, 1661, doi:10.1038/s41598-017-01937-4 (2017).
- 7 Caldas, H. & Herman, G. E. NSDHL, an enzyme involved in cholesterol biosynthesis, traffics through the Golgi and accumulates on ER membranes and on the surface of lipid droplets. *Hum Mol Genet* **12**, 2981-2991, doi:10.1093/hmg/ddg321 (2003).
- 8 Chen, J. *et al.* ACSL4 reprograms fatty acid metabolism in hepatocellular carcinoma via c-Myc/SREBP1 pathway. *Cancer Lett* **502**, 154-165, doi:10.1016/j.canlet.2020.12.019 (2021).
- 9 Zheng, R., Zhu, H. L., Hu, B. R., Ruan, X. J. & Cai, H. J. Identification of APEX2 as an oncogene in liver cancer. *World J Clin Cases* **8**, 2917-2929, doi:10.12998/wjcc.v8.i14.2917 (2020).
- 10 Chang, C. C. *et al.* CTP synthase forms the cytoophidium in human hepatocellular carcinoma. *Exp Cell Res* **361**, 292-299, doi:10.1016/j.yexcr.2017.10.030 (2017).
- 11 Dong, W. *et al.* The RNA-binding protein RBM3 promotes cell proliferation in hepatocellular carcinoma by regulating circular RNA SCD-circRNA 2 production. *EBioMedicine* **45**, 155-167, doi:10.1016/j.ebiom.2019.06.030 (2019).

- 12 Colemonts-Vroninks, H. *et al.* Oxidative Stress, Glutathione Metabolism, and Liver Regeneration Pathways Are Activated in Hereditary Tyrosinemia Type 1 Mice upon Short-Term Nitisinone Discontinuation. *Genes (Basel)* **12**, doi:10.3390/genes12010003 (2020).
- 13 Zhang, Z. *et al.* DHHC9-mediated GLUT1 S-palmitoylation promotes glioblastoma glycolysis and tumorigenesis. *Nat Commun* **12**, 5872, doi:10.1038/s41467-021-26180-4 (2021).
- 14 Lee, S. R. *et al.* Loss of PGRMC1 Delays the Progression of Hepatocellular Carcinoma via Suppression of Pro-Inflammatory Immune Responses. *Cancers (Basel)* **13**, doi:10.3390/cancers13102438 (2021).
- 15 Stanya, K. J. *et al.* Direct control of hepatic glucose production by interleukin-13 in mice. *J Clin Invest* **123**, 261-271, doi:10.1172/JCI64941 (2013).
- 16 Zhang, B. *et al.* KDM5C Represses FASN-Mediated Lipid Metabolism to Exert Tumor Suppressor Activity in Intrahepatic Cholangiocarcinoma. *Front Oncol* **10**, 1025, doi:10.3389/fonc.2020.01025 (2020).
- 17 Wang, J. *et al.* Proteomic Analysis and Functional Characterization of P4-ATPase Phospholipid Flippases from Murine Tissues. *Sci Rep* **8**, 10795, doi:10.1038/s41598-018-29108-z (2018).
- 18 Liu, M. *et al.* Allele-specific imbalance of oxidative stress-induced growth inhibitor 1 associates with progression of hepatocellular carcinoma. *Gastroenterology* **146**, 1084-1096, doi:10.1053/j.gastro.2013.12.041 (2014).
- 19 Chen, G., Wang, H., Xie, S., Ma, J. & Wang, G. STAT1 negatively regulates hepatocellular carcinoma cell proliferation. *Oncol Rep* **29**, 2303-2310, doi:10.3892/or.2013.2398 (2013).

REVIEWERS' COMMENTS:

Reviewer #1 (Remarks to the Author):

Thanks for the adequate reaction and explanation, I don't have further comments.

Reviewer #2 (Remarks to the Author):

Thorough analysis of the effects of maternal obesity in a rodent model. Authors addressed all the comments made by the reviewers previously. The manuscript is now improved and easier to comprehend. I have no further comments to make.